# Visual Decoding and Reconstruction via EEG Embeddings with Guided Diffusion

**Dongyang Li**[1*]   **Chen Wei**[1*]   **Shiying Li**[1]   **Jiachen Zou**[1]   **Quanying Liu**[1†]

[1]Department of Biomedical Engineering, Southern University of Science and Technology, Shenzhen, China

{lidy2023, weic3}@mail.sustech.edu.cn

liuqy@sustech.edu.cn

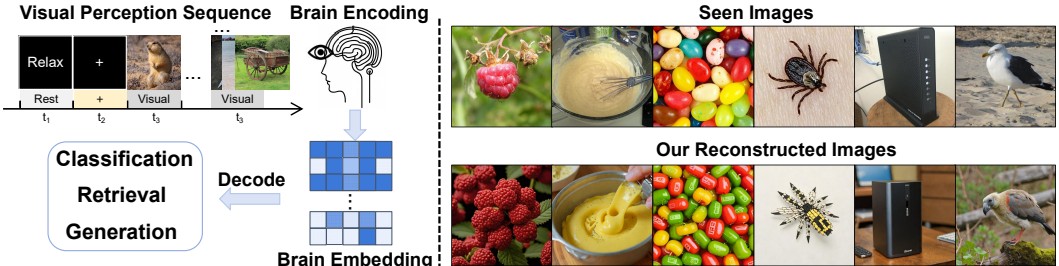

Figure 1: **EEG/MEG-based zero-shot brain decoding and reconstruction.** Left: Overview of three visual decoding tasks using EEG/MEG data under natural image stimulus. Right: Our reconstruction examples.

## Abstract

How to decode human vision through neural signals has attracted a long-standing interest in neuroscience and machine learning. Modern contrastive learning and generative models improved the performance of visual decoding and reconstruction based on functional Magnetic Resonance Imaging (fMRI). However, the high cost and low temporal resolution of fMRI limit their applications in brain-computer interfaces (BCIs), prompting a high need for visual decoding based on electroencephalography (EEG). In this study, we present an end-to-end EEG-based visual reconstruction zero-shot framework, consisting of a tailored *brain encoder*, called the Adaptive Thinking Mapper (ATM), which projects neural signals from different sources into the shared subspace as the clip embedding, and a *two-stage multi-pipe EEG-to-image generation strategy*. In stage one, EEG is embedded to align the high-level clip embedding, and then the prior diffusion model refines EEG embedding into image priors. A blurry image also decoded from EEG for maintaining the low-level feature. In stage two, we input both the high-level clip embedding, the blurry image and caption from EEG latent to a pre-trained diffusion model. Furthermore, we analyzed the impacts of different time windows and brain regions on decoding and reconstruction. The versatility of our framework is demonstrated in the magnetoencephalogram (MEG) data modality. The experimental results indicate that our EEG-based visual zero-shot framework achieves SOTA performance in classification, retrieval and reconstruction, highlighting the portability, low cost, and high temporal resolution of EEG, enabling a wide range of BCI applications. Our code is available at https://github.com/ncclab-sustech/EEG_Image_decode.

---

[*]D. Li and C. Wei contributed equally.

[†]Corresponding author.

38th Conference on Neural Information Processing Systems (NeurIPS 2024).

# 1   Introduction

A key technical challenge in BCIs is to decode/reconstruct the visual world seen by humans through non-invasive brain recordings, such as fMRI, MEG or EEG. These highly dynamic brain activities reflect human perception of the visual world, which is influenced by properties of the external visual stimulus, our internal states, emotions and even personal experiences. Thus, visual decoding and reconstruction based on neural signals can uncover how the human brain processes and interprets natural visual stimulus, as well as promote non-invasive BCI applications.

Contrastive learning and generative models have greatly advanced fMRI-based visual decoding in both decoding tasks (e.g., image classification and retrieval) and generative tasks (e.g., image reconstruction). By combining pre-trained visual models, existing fMRI decoding models can learn highly-refined feature embeddings in limited data [1, 2]. Using these embedded fMRI features, generative models such as diffusion models can reconstruct the image one is seeing [2, 3]. However, despite many advances in fMRI-based visual decoding, fMRI equipment is unportable, expensive, and difficult to operate, largely limiting its application in BCIs. Alternatively, EEG is portable, cheap, and universal, facilitating a wide range of BCI applications. EEG has higher temporal resolution and can effectively capture rapid changes in brain activity when processing complex, dynamic visual stimulus.

EEG has long been considered incomparable to fMRI in natural image decoding/reconstruction tasks, as EEG suffers from low signal-to-noise ratio, low spatial resolution, and large inter-subject variability. Recent advances in multimodal alignment have made MEG/EEG visual decoding possible, although the performance is still inferior to fMRI [4, 5, 6]. Yohann Benchetrit et al. used the CLIP model to extract the latent representation of the image and trained the MEG encoder to align it with the image representation extracted by CLIP. It achieved excellent retrieval and reconstruction performance on MEG and fMRI datasets, demonstrating the potential for real-time visual decoding and reconstruction using EEG/MEG signals. Recently, Song et al. [5] used an EEG encoder based on ShallowNet [7] and performed representation alignment through contrastive learning, achieving excellent decoding performance on the THING-EEG dataset [8]. These two studies provide preliminary evidence of the potential of EEG/MEG-based visual decoding. However, there is a significant gap in their performance compared to the fMRI-level performance. This gap is largely caused by the fact that the framework of EEG visual decoding and reconstruction have not yet been thoroughly explored.

To fill this gap, we have developed a visual decoding framework based on EEG/MEG, including a novel EEG encoder and a two-stage image generation strategy. Our work has three main contributions:

1. We present brain decoding framework, which is the first work allows zero-shot image classification, retrieval, and reconstruction via EEG data. Experimental results demonstrate that our framework is applicable to various common EEG encoder architectures.

2. By extensively studying the existing EEG encoder modules, we construct a tailored EEG encoder ATM, which achieves state-of-the-art performance in three downstream visual decoding tasks.

3. We report a two-stage EEG-to-image generation strategy, which separately extracts high-level and low-level visual features from EEG and refining these features with an additional lightweight prior diffusion model, enabling reliable reconstruction of images using less than 500ms EEG.

# 2   Method

To learn high-quality latent representations of EEG data, it is crucial to consider the spatial position of EEG channels and the Temporal-Spatial properties of EEG signals. Let $T$ represent the length of the time window of the data, $C$ the number of EEG channels, and $N$ the total number of data samples. Our objective is to derive EEG embeddings $Z_E = f(E) \in \mathbb{R}^{N \times F}$ from the brain activity data $E \in \mathbb{R}^{N \times C \times T}$, where $f$ is the EEG encoder and $F$ is the projection dimension of the embeddings. Concurrently, we use the CLIP model to extract image embeddings $Z_I \in \mathbb{R}^{N \times F}$ from images $I$. Our goal is to effectively align the EEG representation with the image representation, as illustrated in Fig. 2. In the training phase, the EEG encoder is trained with EEG and image pairs using a contrastive learning framework. In the inference phase, the EEG embeddings from the trained EEG projector

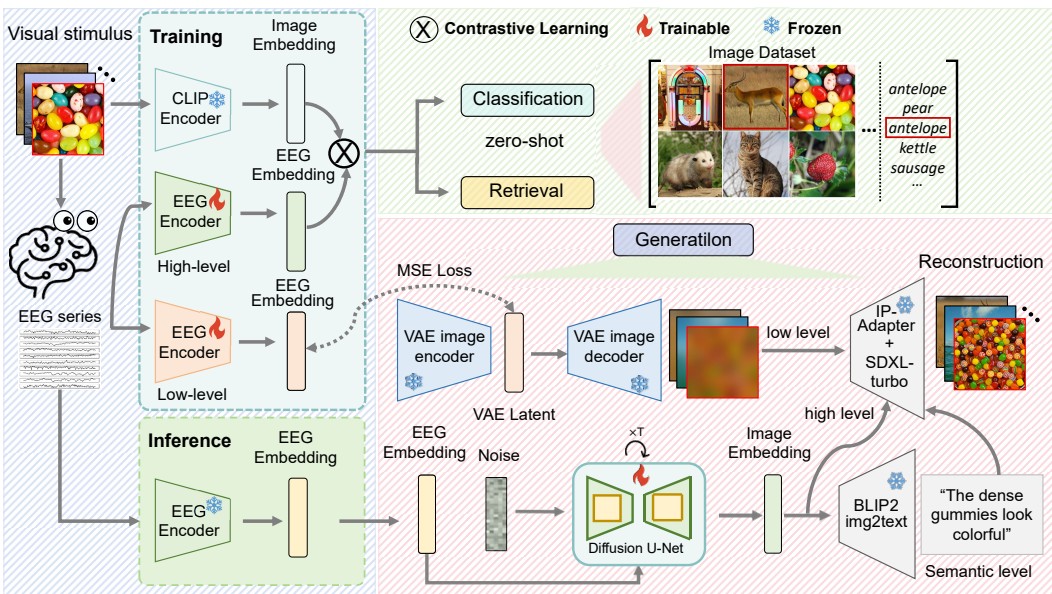

Figure 2: **EEG/MEG-based visual decoding and generation framework.** The EEG encoder is designed as a flexible replacement component. After aligning with image features, the EEG features are used for zero-shot retrieval and classification tasks, and the reconstructed images are obtained through a two-stage generator.

can be used for a variety of zero-shot tasks, including EEG-based image classification, retrieval, and image reconstruction.

## 2.1 ATM for EEG Embedding

Inspired by advanced time series models [9, 10], we develop an EEG encoder called ATM, for aligning the original EEG signals to its feature representation space (Fig. 3). ATM is based on the channel-wise Transformer encoder, Temporal-Spatial convolution and multilayer perceptron (MLP) architecture. In contrast to other conventional practices, the original EEG does not need to be segmented, and each sequence acts as a patch. After sinusoidal position embedding, these patches are processed through a channel-attention module to integrate the information of different series. Subsequently, through the Temporal-Spatial aggregation, we project the output with a MLP to get rational shape representations. The Temporal-Spatial convolution module is an effective way to represent EEG data with a small number of parameters [5], prevent overfitting in training. The difference is our components is plug-and-play and can be flexibly replaced with different types of Temporal-Spatial convolution components as needed to adapt to various EEG/MEG datasets. Finally, MLP projectior consists of $M$ simple residual components and fully connected layers, with LayerNorm applied in the output to ensure the stability of training. In addition to entering the original series, we provide an identification input for a known subject and can specifically use this token for downstream tasks. For unknown subjects, we use shared tokens or average all tokens equally directly into the MLP projector.

## 2.2 Image Embedding

Many previous studies have explored various training strategies to train deep neural networks for image embedding, such as VGG-19 and ResNet trained with supervised learning, CLIP, DINO trained with contrastive learning, and VAEs with self-supervised learning [11, 12, 5, 6]. They have reported that DINO and CLIP models pre-trained using the Vision Transformer (ViT) architecture perform better in a range of downstream tasks, including image decoding and reconstruction, compared to models trained using supervised learning methods (such as VGG, ResNet) and self-supervised VAE frameworks. Thus, in this study, we use CLIP for image embedding, denoted as $Z_I \in \mathbb{R}^{N \times 1024}$, instead of $Z_I \in \mathbb{R}^{N \times 257 \times 768}$, with the EEG embeddings. Before formal training, all images undergo the standard preprocessing procedure [13].

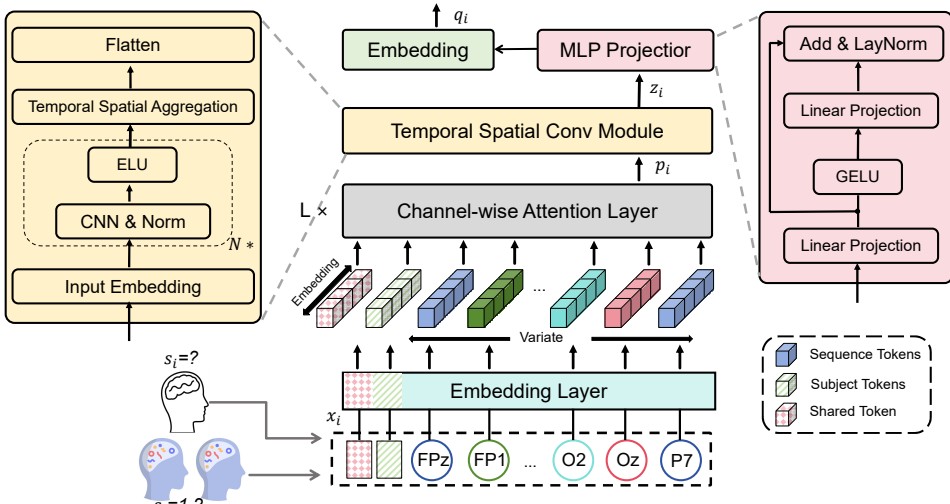

Figure 3: **The structure of ATM**. The original EEG sequences of different variates are independently embedded into tokens. Channel-wise attention is applied to embedded variate tokens with enhanced interpretability revealing electrode correlations. And representations of each token are extracted by the shared feedforward network (FFN). Then Temporal-Spatial convolution can prevent overfitting and enhance the ability of Temporal-Spatial modeling.

## 2.3 EEG Guidance Image Generation

In this study, we present a two-stage pipeline for generating images that serve as visual stimulus for EEG recordings, as shown in the bottom right of Fig. 2. In the left of Fig. 3 we have obtained the EEG embeddings $z_E$ for each image by the EEG encoder ATM. Now our goal is to use these EEG embeddings to generate the corresponding images. The joint distribution of images, EEG embeddings, and image embeddings can be expressed as $p(I, z_E, z_I) = p(z_I|z_E)p(I|z_I)$, corresponding to the prior diffusion and CLIP-guided generation, respectively. In **Stage I**, we first focus on the prior diffusion stage. Inspired by DALL-E 2 [14] and Mind's Eyes [2], we train a diffusion model conditioned on the EEG embeddings $\hat{Z}_E$ to learn the distribution of CLIP embeddings $p(z_I|z_E)$. In this stage, we construct a lightweight U-Net: $\epsilon_{\text{prior}}(z_I^t, t, z_E)$, where $z_I^t$ represents the noisy CLIP embedding at diffusion time step $t$. We train the prior diffusion model using EEG and CLIP embeddings. Through this diffusion model, we can generate corresponding CLIP embeddings $z_I$ from EEG embeddings as a prior for stage II. In **Stage II**, we employ the pre-trained SDXL [15] and IP-Adapter [16] models to model the generator $p(I|z_I)$, thereby sampling image $I$ according to $z_I$. In addition, we introduce the low-level features here using img2img[17]. Further details are provided in Appendix C.

## 2.4 Loss Function

Following the methodology outlined by Benchetrit et al. [12], we adopt a dual approach to loss functions, serving distinct objectives. For the classification and retrieval tasks, we only utilize the CLIP loss, which is inspired by the contrastive learning approach described in Radford et al. [13]. This loss function aids in aligning the EEG data $E$ with corresponding image data $I$, thereby facilitating the identification of EEG-image pairs and maximizing the boundaries of EEG representations. For the generation tasks, besides the CLIP loss, we add a Mean Squared Error (MSE) loss to facilitate consistency learning in regression. Thus the overall loss function for our model is a combination of these two distinct loss types, expressed as:

$$Loss = \lambda \cdot L_{CLIP} + (1 - \lambda) \cdot L_{MSE}$$

Here, $\lambda$ is a hyperparameter that balances the contribution of each loss type.

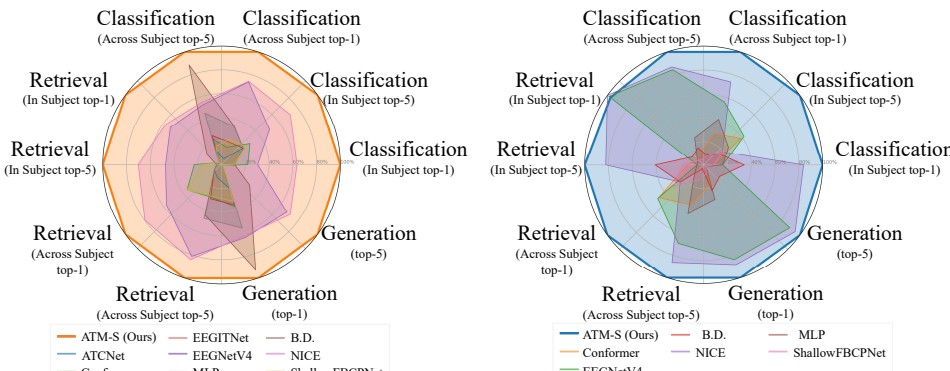

Figure 4: **EEG/MEG-based decoding and reconstruction performance.** Left: Comparisons of nine encoders on the THINGS-EEG dataset, including within-subject and cross-subject performance. Right: Comparisons on the THINGS-MEG dataset, similar to left. Our method achieves the highest performance compared to other competing encoders in EEG/MEG-based visual decoding tasks.

# 3  Experiments

## 3.1  Training and Computational Considerations

We conducted our experiments on the THINGS-EEG dataset's training set [8, 6]. To verify the versatility of ATM for embedding electrophysiological data, we tested it on MEG data modality using the THINGS-MEG dataset [18]. All experiments can be completed in a single NVIDIA RTX 4090 GPU. We used the Adam optimizer [19] to train the across-subject model on a set of approximately 496,200 samples, and the within-subject model on a set of about 66,160 samples, with an initial learning rate of $3 \times 10^{-4}$ and batch sizes of 16 and 1024. Our initial temperature parameter was set to 0.07. We splited the last batch of the original training set as the validation set and selected the best model based on the minimum validation loss over 40 epochs. For fairness, all models' hyperparameters were kept consistent. In our study, we compared the performance of different encoders on the within-subject test set and cross-subject (leave-one-subject-out) test set (see Appendix H).

## 3.2  EEG Decoding Performance

Our method obtains the EEG embedding for the classification task. We output the category of EEG with the highest cosine similarity with text embeddings(Fig. 5a). Fig. 5c presents the average accuracy across different methods in the subjects, and shows that our method outperforms others. More details of the EEG-based image classification are in Appendix B.

In Fig. 5, we test the effectiveness of EEG embeddings in the image retrieval task. We calculate the cosine similarity between the EEG embeddings and the CLIP embeddings instead of text embeddings in the image dataset (with 200 images). Fig. 5d shows the average results for all subjects. We take the highest test accuracy in the evaluation process as the statistical result. Fig. 5b shows the Top-5 retrieved images corresponding to the real visual stimulus seen by subjects. Compared with the previous models, the Top-1 accuracy of our model is significantly improved, and the Top-5 images all maintain a high degree of similarity with the original images. See the Tab. 8 in Appendix for more detailed averages of test accuracy in subjects.

**Ablation Study on ATM**    We systematically deconstructed and analyzed each layer of our EEG projector. We conducted an ablation study for each component in ATM (i.e., the MLP projector, the Temporal-Spatial convolution module and the channel-wise attention block). We specified two different convolution architectures, ShallowNet (ATM-S) and EEGNetV4 (ATM-E), as our convolution backbone. Appendix B.3 showed the results obtained under different experimental configurations.

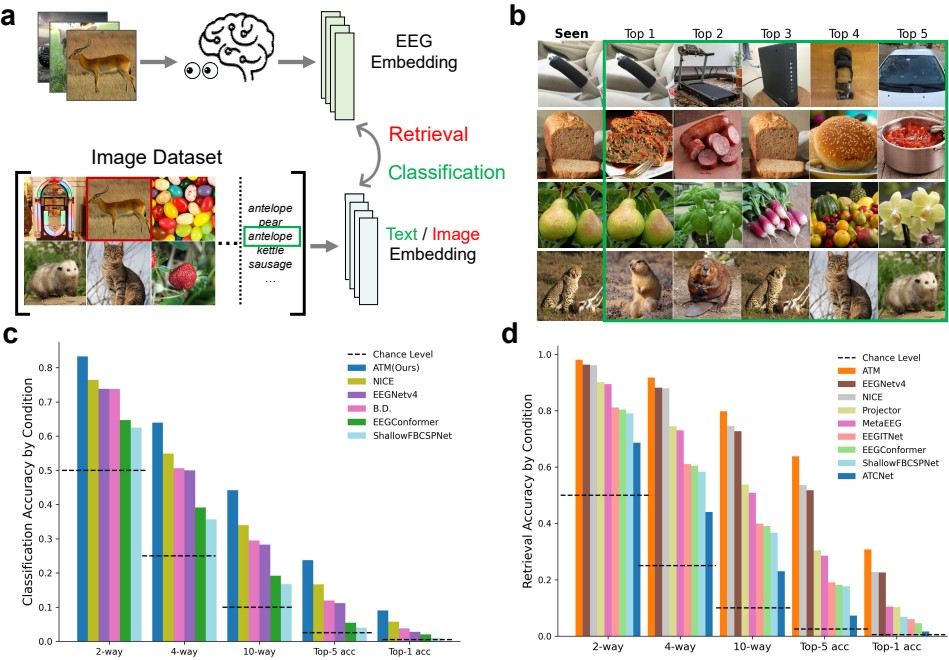

Figure 5: **EEG-based image retrieval and classification**. (a) The paradigm of EEG-based image retrieval and classification. (b) Samples of the top-5 accuracy in EEG-image retrieval tasks. See Appendix G for additional images results. (c) Average in-subject classification accuracy across different methods. (d) Average in-subject retrieval accuracy across different methods.

## 3.3 Image Generation Performance

Fig. 6a shows the process of generating images under the guidance of EEG embedding and evaluating the quality of the generated images. To evaluate the generation performance, we conducted an image retrieval task. Specifically, we extract the CLIP embedding of the generated images and compare the similarity between the CLIP embeddings of all images to retrieve the generated image.

Fig. 6b shows the similarity of distribution. Fig. 6c shows the generated samples. The generated images have high semantic similarity with the seen images and have good diversity in low-level visual features, which can be manipulated by the guidance scale hyperparameter (Fig. 6d). We also report the reconstruction performance for EEG, MEG, and fMRI across various metrics from different methods and datasets in the Tab. 1.

Table 1: Quantitative assessments of the reconstruction quality for EEG, MEG, and fMRI in Subject 8. For detailed explanations of the metrics.

| Dataset | Low-level | | | High-level | | | |
|---|---|---|---|---|---|---|---|
| | PixCorr ↑ | SSIM ↑ | AlexNet(2) ↑ | AlexNet(5) ↑ | Inception ↑ | CLIP ↑ | SwAV ↓ |
| NSD-fMRI [12] | 0.305 | 0.366 | 0.962 | 0.977 | 0.910 | 0.917 | 0.410 |
| NSD-fMRI [20] | 0.254 | 0.356 | 0.942 | 0.962 | 0.872 | 0.915 | 0.423 |
| NSD-fMRI [21] | 0.130 | 0.308 | 0.917 | 0.974 | 0.936 | 0.942 | 0.369 |
| THINGS-MEG [12] | 0.058 | 0.327 | 0.695 | 0.753 | 0.593 | 0.700 | 0.630 |
| THINGS-MEG (averaged) [12] | 0.076 | 0.336 | 0.736 | 0.826 | 0.671 | 0.767 | 0.584 |
| THINGS-MEG (Ours) | 0.104 | 0.340 | 0.613 | 0.672 | 0.619 | 0.603 | 0.651 |
| **THINGS-EEG (Ours)** | **0.160** | **0.345** | **0.776** | **0.866** | **0.734** | **0.786** | **0.582** |

## 3.4 Temporal Analysis

To investigate the effects of different EEG time window on visual decoding, we calculated the average top-1 classification accuracy for sliding and growing time windows: $[0, t]$, including the entire period

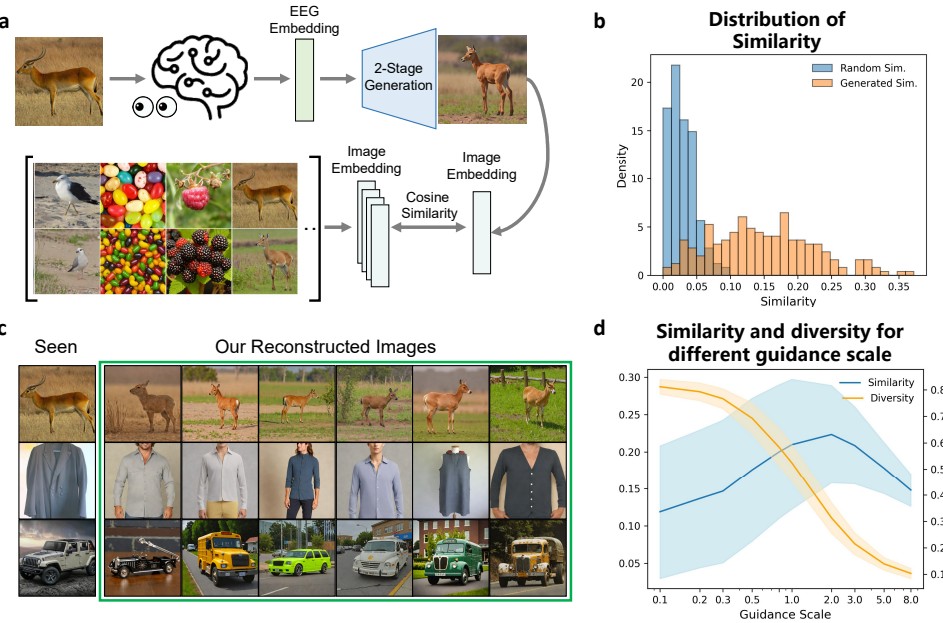

Figure 6: **EEG guidance image generation**. (a) The paradigm of image generation. (b) The similarity between random visual objects and the EEG embeddings, and the similarity between generated visual objects and the target EEG embeddings. (c) Comparison between the original image and the image generated using the corresponding EEG data. (see Appendix C for details). (d) The similarity between visual objects and target EEG embeddings as the guidance scale changes, and the diversity of visual objects as the guidance scale changes. See Appendix G for additional results.

from the onset of visual stimulus to time point $t$, and [$t$-100, $t$], only including the data 100ms before time point $t$. We compared the accuracy with a randomly selected baseline (0.5% chance level) to test predictive performance (Fig. 7). Our results show that within 500ms after visual stimulus, the accuracy reaches an upper limit of about 30%, after which the accuracy no longer improves (Fig. 7a). The MEG decoding shows a similar profile as the time window expands (Fig. 7b). We exhibit the generated images under different EEG time windows, [0, $t$] in Fig. 7c. The similarity is low when the time window is less than 150ms, and this similarity gradually increase as the time window expands. After 500 milliseconds, EEG-guided image generation can reliably reveal the semantics of the images seen. Interestingly, we find differences in the optimal reconstruction time windows for different categories of images, for example, jelly beans (200ms) are faster than aircraft carrier (500ms), implying that the human brain may process different visual objects at different speeds. This finding highlights the advantage of EEG's high temporal resolution in studying fast visual processing compared with the lower temporal resolution of fMRI.

### 3.5 Spatial Analysis

To investigate the contribution of different brain regions to visual decoding, we divided the EEG electrodes from the THING-EEG data into five distinct brain regions (i.e., Frontal, Temporal, Center, Parietal, Occipital regions in Fig. 8a), and then conducted ablation experiments on retrieval task (Fig. 8b) and the reconstruction task (Fig. 8c). The results showed that using information from all brain regions is optimal, for both retrieval and generation tasks. The occipital had the highest retrieval accuracy and reconstruction performance compared to other regions. Parietal and temporal regions contain some semantic information, whereas frontal and central regions contribute the least useful information to the visual decoding.

## 4 Related Works

**Visual decoding using neural signals:** Decoding visual information from our brain has been a long-standing pursuit in neuroscience and computer science [22, 23]. Some progress has been

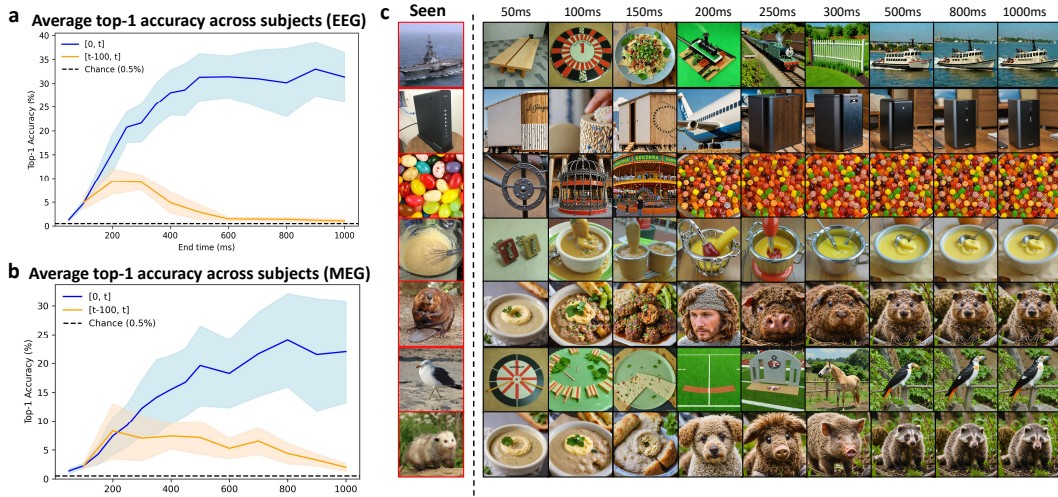

Figure 7: **Performance of different EEG/MEG time windows on EEG-guided visual retrieval and reconstruction**. (a) The retrieval accuracy of the expanding EEG windows at intervals [0, t] and at intervals [t-100, t] respectively. (b) The retrieval accuracy of the expanding MEG windows. (c) Samples reconstructed as the EEG window expands. When the EEG time window is greater than 200ms, the reconstructed image is stable. See Appendix H for more detailed explanations.

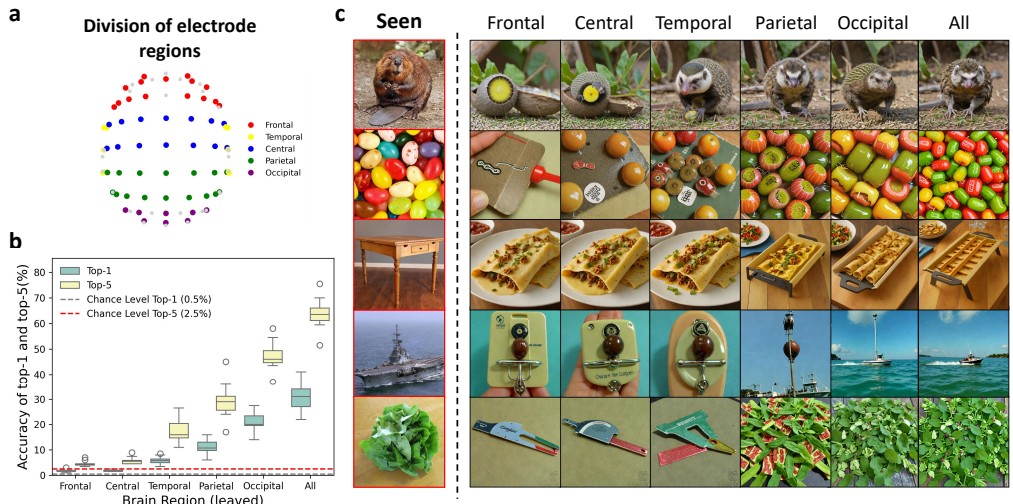

Figure 8: **EEG-guided retrieval and reconstruction using EEG from different brain regions**. (a) The EEG electrodes assigned to five brain regions. (b) Top-1 and top-5 retrieval accuracy, using only the EEG channels in each leaved region and all channels. (c) Reconstructed images obtained using only the electrode channels in each individual region and all channels.

made in decoding steady-state visual stimulus. However, accurately and rapidly decoding semantic information in natural images remains a challenge [24]. fMRI has been widely used to estimate semantic and shape information in visual processing within the brain [25, 26]. However, the demand for high-speed and practical applications in brain-computer interfaces calls for alternative approaches. EEG, due to its high temporal resolution and portability, emerges as a promising option [27]. Yet, the overall performance across different subjects and biological plausibility remains unresolved [28]. Furthermore, previous approaches often relied on supervised learning methods with limited image categories, overlooking the intrinsic relationship between image stimulus and brain responses [1, 29, 30].

**Neural decoding using EEG/MEG data:** Previous studies have shown the efficacy of Temporal-Spatial modules in representing neural data [7, 31]. For example, lightweight convolutional neural

networks such as EEGNet and ShallowNet [7] have achieved considerable performance in small EEG and MEG datasets. Using contrastive learning, it has been shown that merely using convolutional neural networks and projection layers can yield satisfactory results on neural datasets [32]. More recently, Benchetrit et al proposed a method towards real-time MEG-based reconstruction of visual perception [12]. Song et al. presented an EEG encoder using ShallowNet Temporal-Spatial convolution module with a large convolution kernel with a few parameters for EEG embedding, resulting in favorable performance on EEG-based visual decoding [5].

**Limitations of previous studies:** Previous EEG studies are primarily oriented toward understanding visual perception in the human brain rather than maximizing EEG decoding performance. Thus the visual decoding performance is far from optimal. Specifically, previous studies have trained linear models to (1) classify a small set of images from brain activity [33, 34], (2) to predict brain activity from the latent representations of images [4], or (3) to quantify the similarity analysis between these two patterns with representational similarity [4, 33, 8, 35]. While these studies also utilize image embeddings, their linear decoders are limited to classifying a small group of object categories or distinguishing image pairs. Moreover, several deep neural networks have been applied to maximize classification of speech [36], cognitive load [37], and images [38, 39, 40] in EEG recordings. [38] proposed a deep convolutional neural network for classifying natural images using EEG signals. Unfortunately, the experiment presented all images of the same category in a single block, probably misleading the decoder to rely on autocorrelated noise rather than the hidden informative patterns of brain activity [30]. Also, these EEG studies only classify a relatively small number of image categories.

## 5    Discussion and Conclusion

In this study, we proposed a novel and feasible EEG-based zero-shot image reconstruction framework. Although it utilizes existing machine learning techniques , we demonstrate for the first time that EEG-based zero-shot visual decoding and reconstruction can be competitive with MEG and fMRI.

**Technical Impact:** Our technical contributions are mainly on the EEG encoder and the two-stage zero-shot visual reconstruction framework (Fig. 2). First, we developed the ATM, an EEG encoder which can efficiently represent EEG/MEG features for three tasks. Our comprehensive experiments of the EEG encoder (Fig. 3), compared to various architectures and training methods, achieves SOTA performance across various metrics and tasks (Figs. 4b, 5). Second, our two-stage EEG guidance image reconstruction framework achieves performance close to fMRI using only EEG data (Figs. 6, 14, Tab. 1, 6), and this method is compatible with MEG data (Figs. 4c, 7b).

**Neuroscience Insights:** Our results offer insights into the relationship between brain activity and visual perception. We analyzed EEG-based visual decoding within different time windows to examine when visual information is perceived in the brain (Fig. 7). Our results revealed that visual information in EEG data is predominantly contained within the 200-400ms range (Fig. 7a), consistent with previous EEG studies [11, 6, 5]. Interestingly, the visual information in MEG data last up to 800ms, much longer than EEG (Fig. 7b), in line with the results reported by a previous MEG study [12, 5]. We also found that EEG performs better than MEG in visual tasks (See Appendix D for Tab. 6), which is different from other fields, such as speech decoding [36]. In addition, through ablation experiments of spatial information, we found that visual information is mainly encoded in the occipital and parietal areas (Fig. 8).

**Interesting Phenomena and Future Directions:** First, there are non-negligible performance differences between cross-subject and within-subject settings. This performance gap arises from inter-subject differences in EEG signals [41, 42], likely attribute to heterogeneity in individual brain, differences in visual perception between individuals, and even shifts in noise distribution during EEG recording. So it calls for more efforts on EEG encoder, such as more flexible neural network architectures or better weight initialization of pre-trained models [43, 44]. Transfer learning and meta-learning are also future directions worth exploring [45, 46, 47]. Moreover, how to unify various electrode montages of different EEG datasets when pre-training large EEG models is a challenge. EEG source localization, which converts senor-level EEG signals into the standard brain source space [48, 49], might be a potential solution.

## Acknowledgements

This work is supported by the National Key R&D Program of China (2021YFF1200804), Shenzhen Science and Technology Innovation Committee (202009251559957004, KCXFZ20201221173400001, SGDX2020110309280100).

This work originated as a course project in the BI&AI 2023 course at SUSTech, and the author would like to thank course project members, Yuanhao Fan and Zhihong Wu, for their contributions to the early stages of this work. Thank Haoyang Qin for his time and advice.

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

# Supplementary Material:

# Visual Decoding and Reconstruction via EEG Embeddings with Guided Diffusion

## A  Datasets for experiments

### A.1  EEG dataset

We conducted our experiments on the THINGS-EEG dataset's training set [8, 6]. This dataset includes a large EEG corpus from 10 human subjects during the visual task. The experiment employed the Rapid Serial Visual Presentation (RSVP) paradigm for orthogonal target detection tasks to ensure that participants attended to the visual stimulus. All 10 participants completed 4 equivalent experiments, resulting in 10 datasets with 16,540 training image conditions repeated 4 times, and 200 testing image conditions repeated 80 times, totaling (16,540 training image conditions $\times$ 4 repetitions) + (200 testing image conditions $\times$ 80 repetitions) = 82,160 image trials. Original data were collected using a 64-channel system at a sampling rate of 1000 Hz. After signal denoising, epoch data were downsampled to 100 Hz, selecting 17 channels covering the occipital and parietal cortex. Instead of using the raw dataset, we chose to filter it to [0.1, 100] Hz, retaining 63 channels of the original EEG data at a sampling rate of 1000 Hz. For preprocessing, we segmented the EEG data from 0 to 1000 ms after the stimulus onset into trials. Baseline correction was performed using the mean of the 200 ms pre-stimulus data. All electrodes were retained and downsampled to 250 Hz for analysis, and multivariate noise normalization was applied to the training data [50]. To improve signal-to-noise ratio, we averaged across the 80 EEG trials from the same image in the test set, while keeping each EEG trial in the training setting. We compared the effects of averaging across EEG trials and found it indeed improved the performance.

### A.2  MEG dataset

To verify the versatility of ATM for embedding electrophysiological data, we tested it on MEG data modality using the THINGS-MEG dataset [18]. It includes 271-channel MEG data from 4 subjects with 12 MEG sessions. The training dataset has 1854 Concepts $\times$ 12 images $\times$ 1 repetition, and the test dataset has concepts $\times$ 1 image $\times$ 12 repetitions for 200 times. Here, we discarded 200 testing concepts from the training set to construct the same zero-shot task as with the THINGS-EEG. Each image in the THINGS-MEG was displayed for 500 ms. There was a fixed time for each image of $1000 \pm 200$ ms. Continuous MEG data from -100 ms to 1300 ms was segmented into trials after the stimulus onset from 0 to 1000 ms. Preprocessing was performed using a bandpass filter of [0.1, 40] Hz and baseline correction after downsampling to 200 Hz. Note that due to the small number of participants, no statistical analysis was performed on the MEG dataset. We compared our approach with advanced methods i.e. NICE [5] and B.D. [12] for classification and retrieval tasks on the MEG dataset. We directly used the stimulus images to match the template, rather than other images belonging to the concept.

## B  More Implementation Details

### B.1  Evaluation metric implementation

**Classification accuracy**    As CLIP has been designed to align text and image modalities, we also leverage its text encoder for EEG classification using the text embeddings of categories. This approach utilizes CLIP's text encoding capabilities to facilitate EEG classification. We conducted zero-shot classification tests on the THINGS-EEG dataset. We employed **Top-K accuracy** as a metric for performance evaluation. Specifically, we assessed performance based on the Top-k (where k=1, 5) predictions. We conducted tests for both within-subject and leave-one-subject-out classification accuracy, enabling a comprehensive evaluation of the model's performance across different scenarios. Additionally, for each test instance, we extracted embeddings of N-1 unrelated samples from the test set as inputs. This means, apart from the entire test set, the model evaluated by **N-Way accuracy** (where N=2, 4, 10 in our experiments) on the test set. We report these results in Appendix H.

**Retrieval accuracy**   Similar to the classification task, in the retrieval task, the objective is to retrieve the Top-K images most related to a given stimulus image via its corresponding EEG signal. This implies that by changing the text embeddings of image labels to image embeddings, we can transition the task from classification to image retrieval. Given that contrastive learning is known to be sensitive to batch size, we also compared the performance improvement of different methods under varying batch sizes (batch size=16, 1024) (Appendix H).

**Generation accuracy**   The generation task presents more challenges than the other tasks. For each image condition in the test set, we generate 10 different images from 10 subjects based on the corresponding EEG. Subsequently, image retrieval is performed for each generated image. The Top-1 and Top-5 accuracies are calculated. It helps in evaluating the semantic alignment between the generated images and their original counterparts.

## B.2   Computing methods implementation

In the upstream EEG encoder part, we compared various methods. For the B.D. method [12], we replicated the network structure as described in the original work, with the difference being in the shape of the input data due to the original study's focus on MEG. It is worth mentioning that we used the leave one for subject method in the testing process so we modify its subject-wise layer as an linear layer for modeling the time dimension. To ensure fairness, we did not use the same hyperparameters as in the original paper. Instead, we chose settings yielded excellent results upon reproduction. Across all methods, we used identical hyperparameters, apart from the network structures. These included batch size, optimizer, initial learning rate, and temperature parameters.

## B.3   Architecture details

Table 2: Brain module configuration

| Layer | Input shape | Output shape | # parameters |
|---|---|---|---|
| Channel-wise attention layer | ($N, C, T$) | ($N, C, D$) | 553,078 |
| Temporal-Spatial Conv module | ($N, C, D$) | ($N, H1, H2$) | 103,680 |
| Temporal-Spatial aggregation | ($N, H1, H2$) | ($N, H1*H2$) | 0 |
| MLP projector | ($N, H1*H2$) | ($N, 1024$) | 2,527,232 |
| **Total** | | | **3,183,990** |

Table 3: Ablation study on the ATM model's different components for THINGS-EEG retrieval.

| Module | MLP | TSConv | CAL | TOP-1 | TOP-5 |
|---|---|---|---|---|---|
|       | ✓ | ✗ | ✗ | 8.11±1.74 | 26.83±4.78 |
|       | ✓ | ✓ | ✗ | 21.65±6.22 | 51.34±9.83 |
| ATM-S | ✓ | ✗ | ✓ | 23.73±7.62 | 52.71 ±9.71 |
|       | ✓ | ✓ | ✓ | **28.64±6.39** | **58.47±8.97** |
|       | ✓ | ✗ | ✗ | 8.11±1.74 | 26.83±4.78 |
|       | ✓ | ✓ | ✗ | 17.95±5.95 | 43.10±8.63 |
| ATM-E | ✓ | ✗ | ✓ | 23.73±7.62 | 52.71 ±9.71 |
|       | ✓ | ✓ | ✓ | **24.92±6.00** | **54.78±8.49** |

## B.4   Model configuration

To validate the efficacy of our EEG encoder, we experimented with a variety of empirical setups aimed at optimizing the model's efficiency. we leverage joint subject training to adapt to new subjects. Once a model is trained, it can be used for both reasoning about known subjects (subject-specific tokens) and reasoning about unknown subjects (shared tokens). In the context of channel-wise attention layer, we explored four distinct approaches for enhancing downstream retrieval capabilities: leveraging subject-specific tokens for retrieval, averaging all tokens for a more generalized retrieval,

flattening the entire token set for direct retrieval, and preserving token dimensions to feed into the Temporal-Spatial convolution module for feature integration. Notably, the strategy of preserving dimensions and using the Temporal-Spatial convolution module emerged as the most effective.

In our quest to optimize token embeddings, we experimented with a variety of approaches, including using $1 \times 1$ convolution and linear layers. Under our framework, using linear layers performs better than convolutions. Moving beyond, we delved into the efficacy of diverse Feed Forward Networks (FFNs) within the Transformer encoder layer. Our findings indicated that an FFN tailored to the temporal dimension emerged as the superior option. We also conducted a comparative analysis of various positional encoding techniques. Interestingly, for Temporal-Spatial convolution utilized in retrieval tasks, the significance of positional encoding diminished.

In a further exploration, we examined the impact of deploying convolutions at various stages and even contemplated the complete removal of the convolution module. It was discovered that situating convolutions post the Transformer encoder layer yielded the most favorable outcomes. Conversely, a shift to a MLP or the removal of the convolution module led to a notable degradation in performance. Our assumption is though the convolution's inherent translational invariance is compromised when the context of the time dimension is disrupted, the efficiency of parameters it possesses may confer a resistance to overfitting, thereby maintaining its effectiveness.

Table 4: Impact of each module on the result in different configurations. The reported results represent the mean performance metrics of the **ATM-S**, calculated over the final 10 training epochs across all 10 subjects.

| Module | Config | Top-1 (std) | Top-5 (std) |
|---|---|---|---|
| Channel-wise attention layer | w/ mean token | 7.29 (3.01) | 23.11 (6.62) |
| | w/ flatten token | 14.85 (5.46) | 37.56 (8.92) |
| | w/ keep dim | **28.64 (6.39)** | **58.47 (8.97)** |
| Token embedding | w/ conv1d | 24.81 (7.29) | 55.68 (9.08) |
| | w/ linear | **28.64 (6.39)** | **58.47 (8.97)** |
| Feed Forward Network | w/ temporal dim | **28.64 (6.39)** | **58.47 (8.97)** |
| | w/ spatial dim | 19.92 (7.74) | 47.59 (11.5) |
| | w/o | 26.45 (7.73) | 57.00 (9.95) |
| Position encoding | w/ sinusoidal | 27.96 (6.54) | 58.16 (8.44) |
| | w/ learnable absolute | 26.66 (6.56) | 56.95 (7.95) |
| | w/o | **28.64 (6.39)** | **58.47 (8.97)** |
| Temporal spatial convolution | w/ pre | 14.23 (4.20) | 37.44 (6.06) |
| | w/ post | **28.64 (6.39)** | **58.47 (8.97)** |
| | w/ both | 25.37 (5.16) | 57.07 (6.13) |
| | w/o | 23.73 (7.62) | 52.71 (9.71) |

### B.5 Training details

In our EEG projector module, we integrated two distinct strategies for steering model predictions: text embedding and image embedding. Given the variance in feature granularity, we observed that alignments that prioritize image embedding excel in tasks of image retrieval and classification. Throughout the training phase, our experiments revealed that a batch size of 16 is a judicious selection for all models. Conversely, a batch size of 1024, which implies a substantial number of samples are processed in each training iteration, necessitates the model to exhibit a heightened capacity for noise resistance. In order to enhance the signal-to-noise ratio within EEG data, we implemented an averaging technique on 80 repeated instances within the test set. This approach mirrors the methodology employed in identifying Event-Related Potentials (ERP). To maximize the utilization of the available training data, we refrained from averaging the 4 repetitions in the training set. Instead, we opted to input the complete set of EEG data into the model, thereby facilitating comprehensive learning.

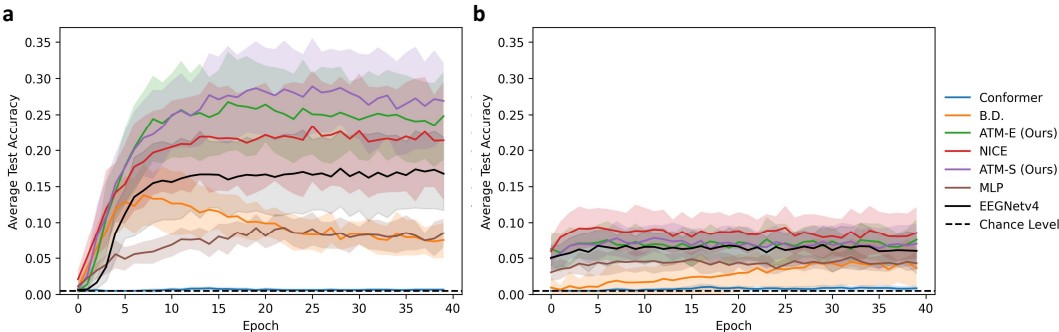

Figure 9: **Test accuracy during each training epoch**. (a) Training of different within-subject models. (b) Training of different across-subject models. We compared 7 different EEG encoding models, including EEGconformer, MLP, EEGNetv4, B.D., NICE, ATM-S (Ours) and ATM-E (Ours).

## C  Details of EEG guidance image generation

Here, we provide a concise overview of the conditional diffusion model framework used in EEG-guided image generation, following the presentation of continuous-time diffusion models in [51, 52].

**Diffusion models**   Diffusion Models (DMs) engage in a generative process by transforming high-variance Gaussian noise into structured data representations. This transformation is achieved by gradually reducing noise levels across a sequence of steps. Specifically, we begin with a high-variance Gaussian noise $x_M \sim \mathcal{N}(0, \sigma_{\max}^2)$ and systematically denoise it through a series of steps to obtain $x_t \sim p(x_t; t)$, where $\sigma_t < \sigma_{t+1}$ and $\sigma_M = \sigma_{\max}$. For a well-calibrated DM, and with $\sigma_0 = 0$, the final $x_0$ aligns with the original data distribution.

**Sampling process**   The sampling in DMs is implemented by numerically simulating a Probability Flow ordinary differential equation (ODE) or a stochastic differential equation (SDE). The ODE is represented as:

$$dx = -\dot{\sigma}(t)\sigma(t)\nabla_x \log p(x; t)dt, \tag{1}$$

where $\nabla_x \log p(x; t)$ is the score function, and $\sigma(t)$ is a pre-defined schedule with its time derivative $\dot{\sigma}(t)$. The SDE variant includes a Langevin diffusion component and is expressed as:

$$
\begin{aligned}
dx = {}& -\dot{\sigma}(t)\sigma(t)\nabla_x \log p(x; t)dt \\
& - \beta(t)\sigma^2(t)\nabla_x \log p(x; t)dt \\
& + \sqrt{2\beta(t)}\sigma(t)d\omega_t,
\end{aligned}
\tag{2}
$$

where $d\omega_t$ is the standard Wiener process.

**Training of DMs**   The core of DM training is to learn a model for the score function. This is typically achieved through denoising score matching (DSM), where $\epsilon_\theta$ is a learnable denoiser. The training process can be formulated as:

$$\mathbb{E}_{(x_0,c)\sim p_{\text{data}}(x_0,c),(n_t,t)\sim p(n_t,t)} \left[ \|\epsilon_\theta(x_0 + \sigma_t\epsilon; t, c) - \epsilon\|_2^2 \right], \tag{3}$$

where $\epsilon$ is Gaussian noise with variance $\sigma_t^2$, and $c$ represents a condition.

### C.1  Stage I - EEG-Conditioned Diffusion

The initiation of the EEG-conditioned diffusion phase is paramount in our EEG-based image generation framework, leveraging the classifier-free guidance strategy alongside data pairs of CLIP embeddings and EEG embeddings $(z_I, z_E)$. Adapting from state-of-the-art generative techniques, our diffusion process is specifically conditioned on the EEG embedding $z_E$ to adeptly capture the distribution of CLIP embeddings $p(z_I|z_E)$. The CLIP embedding $z_I$, procured during this phase, establishes the groundwork for the ensuing image generation stage. Our architecture incorporates a streamlined U-Net, labeled as $\epsilon_{\text{prior}}(z_I^t, t, z_E)$, where $z_I^t$ signifies the perturbed CLIP embedding at a

given diffusion timestep $t$. The training utilizes pairs from the ImageNet database, consisting of over a million images, to fine-tune the EEG-Conditioned Diffusion model. This model is meticulously trained using the classifier-free guidance approach, effectively balancing the conditioning signal's fidelity with the generative output's diversity.

**Classifier-free guidance method** The Classifier-Free Guidance technique is crucial in guiding the iterative refinement of a Diffusion Model (DM) under a specific EEG condition $z_E$. It achieves this by synchronizing the outputs of both a conditional and an unconditional model. The model's formulation, $\epsilon^w_{\text{prior}}(z^t_I; t, z_E)$, is as follows:

$$\epsilon^w_{\text{prior}}(z^t_I; t, z_E) = (1 + w)\epsilon_{\text{prior}}(z^t_I; t, z_E) - w\epsilon_{\text{prior}}(z^t_I; t), \tag{4}$$

where $w \geq 0$ represents the *guidance scale*. This mechanism facilitates concurrent training of the conditional and unconditional models within a singular network framework, periodically substituting the EEG embedding $z_E$ with a null vector to promote training variability, i.e. 10% of the time. The primary objective of this method is to enhance the sample quality produced by DMs while maintaining output diversity.

## C.2 Stage II - CLIP-Embedded Image Synthesis

In Fig. 10, we compare the effects of one-stage and two-stage EEG-guided image generation. We show images generated using EEG embeddings directly (**One-stage**) and images generated using image embeddings obtained via prior diffusion (**Two-stage**). It can be seen that the two-stage EEG-guided image generation can more accurately reconstruct the semantic and low-level visual features of the original image, and the style is more realistic.

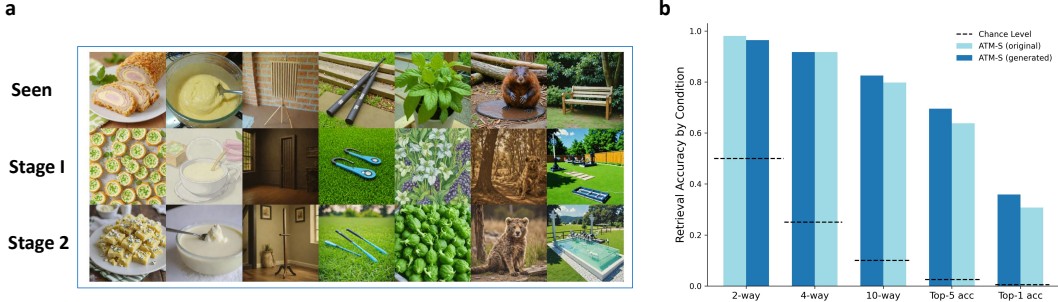

Figure 10: **Comparison between one-stage and two-stage EEG guidance image reconstruction**. (a) We present the images that subjects seen (**Seen**), our reconstructed images directly using EEG embeddings (**One-stage**), and the reconstructed images from low level and high level image embeddings obtained by the prior diffusion (**Two-stage**). These results indicate that the strategy of our two-stage generation can better reconstruct the seen visual stimulus. (b) We employed ATM-S to compare the generated images with the original images in a retrieval task. Our result indicates that the images generated in two stages significantly enhance the performance of the original model on the retrieval task.

In the second stage of our EEG-based image generation approach, the CLIP embedding $z_I$ derived from the EEG-conditioned diffusion acts as the precursor for synthesizing visual objects $I$ based on $z_I$. This is achieved by harnessing the synergies of advanced pre-trained models, namely SDXL and IP-Adapter [15, 16], facilitating the creation of high-caliber images.

The cornerstone of our synthesis process is the SDXL framework, acclaimed for its proficiency in text-to-image conversion. The integration of the IP-Adapter introduces dual cross-attention mechanisms, allowing the CLIP embedding $z_I$ to serve as a directive input and guide the denoising trajectory within the U-Net structure. The synthesis model is denoted as $\epsilon_{\text{SD}}(z_t, t, z_I)$, where $z_t$ denotes the SDXL Variational Autoencoder's (VAE) disturbed latents.

**SDXL-turbo for accelerated processing** To augment the efficiency of our framework, we additionally explore the SDXL-Turbo [53], a refined iteration of SDXL optimized for swift image synthesis. This variant proves especially beneficial in scenarios demanding quick generation of high-fidelity visuals.

**IP-Adapter's efficacy**   The IP-Adapter, with its compact design, has proven to be effective in enhancing image prompt adaptability within pre-trained text-to-image models. Its compatibility with text prompts for multimodal image generation extends the versatility of our EEG-based image synthesis approach.

## C.3   Low-level pipeline

Compared with pure vision pre-training models such as (ViT, ResNet, DINO, etc.), the CLIP model lacks low-level visual features. Therefore, in order to make up for this shortcoming, our framework introduces a low-level visual reconstruction pipeline. We hope to restore basic such as contour, posture, orientation and other pixel-level information from EEG by aligning with the latent of VAE.

Past work [25] has found that in the denoising stage of the early diffusion model, $z$ signals (corresponding to the VAE latent in our framework) dominated prediction of fMRI signals. And during the middle step of the denoising process, $z_c$ predicted activity within higher visual cortex much better than $z$. However, note that this is only an analysis based on decoding accuracy. These analyses do not have a strong neuroscience causal relationship. We still cannot conclude that the low-level features of neural data are modeled by VAE.

Table 5: Latent VAE retrieval performance

| Condition | Top-1 (%) | Top-5 (%) |
|-----------|-----------|-----------|
| ideal chance | 0.5 | 2.5 |
| ATM-S (Ours) | 10.14 | 29.55 |

We trained the low-level pipe for 200 epochs, trying a latent mean squared error (MSE) loss, along with a contrastive learning loss, and a variational autoencoder (VAE) image reconstruction loss to align the $4 \times 64 \times 64$ EEG latents obtained from a projection layer and an upsampled CNN with the VAE latents. Nevertheless, reconstruction loss or contrastive learning loss performs worse than only applying the loss in latent space and also requires significantly more GPU memory. In addition, we found that using a low-level visual model for distillation learning in the low-level pipeline is not only unhelpful for VAE latent training, but also leads to overfitting. Similar conclusions were reached in MindEye[2]. Our results suggested that the low-level zeroshot reconstruction in EEG is not stable enough and may mislead the model results. When using the low level pipeline, we usually set the inference steps of SDXL to 10 (or SDXL-turbo to 4) and the image-to-image denoising strength to 0.5. We give several reconstruction examples in Fig. 11 to compare the impact before and after using low level pipeline. Moreover, low level alignment was validated through retrieval performance tests, as depicted in Tab. 5. Our findings indicate that retrieval using EEG latents can also achieve excellent performance. This further substantiates the feasibility of aligning the low-level consistency through VAE latents.

## C.4   Semantic-level pipeline

In addition to using EEG latent and low-level pipelines during reconstruction, we also add a corresponding semantic level pipeline guided by text captions during the image reconstruction. We input the $1 \times 1024$ EEG features output by prior Diffusion into the trained image projector to obtain $256 \times 1024$ image features. Using the GIT model[54], we can directly generate a caption from the latent features of the image. IP-Adapter accepts such a caption as a text prompt to guide the semantic level reconstruction of the image. It should be noted that due to the difficulty of the zeroshot task itself and the low dimensionality of the EEG features, the caption generated from the latent may be unstable, thereby interfering with the original correct EEG semantics. Considering that the image features extracted by the CLIP model itself are already high-level visual features and do not require the introduction of more semantic information, this framework retains the entry of the text prompt, and the reconstructed image presented does not force the use of the semantic level pipeline.

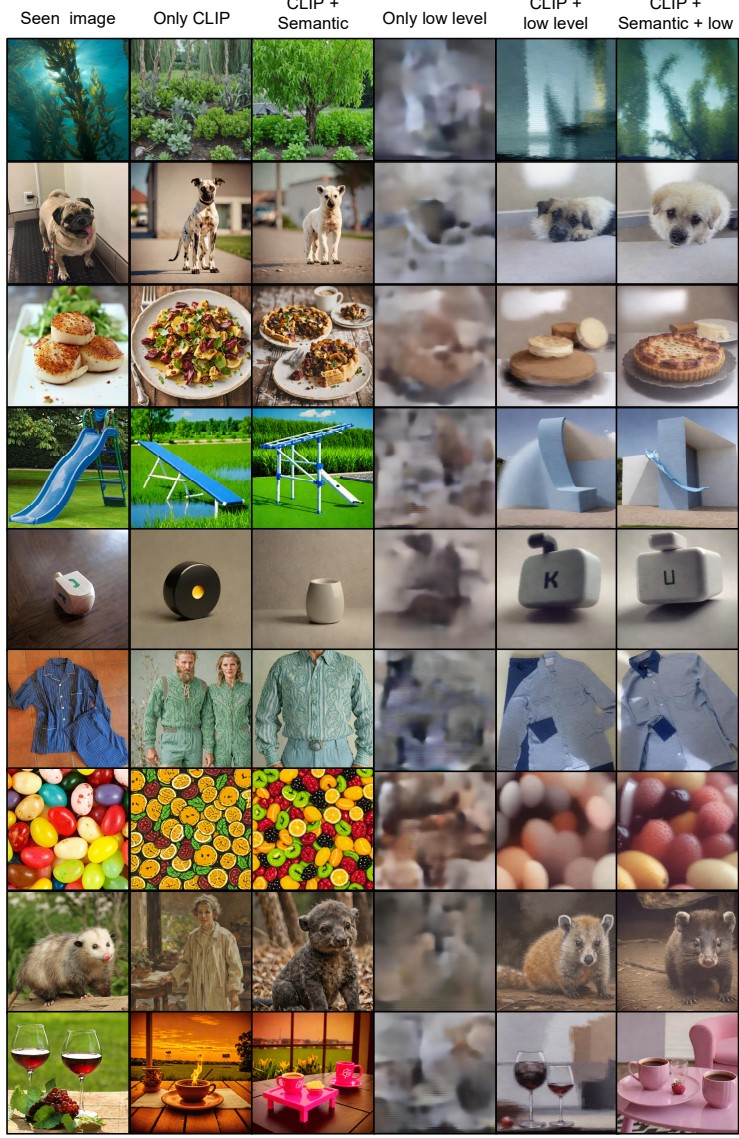

Figure 11: **Example of our reconstructions for Subject 8 output from different pipelines.** From left to right: reconstruction using only CLIP (i.e., Only CLIP), using only CLIP and the semantic pipeline (i.e., CLIP + Semantic), using only the low level pipeline (i.e., Only low level), using only CLIP and the low level pipeline (i.e., CLIP + low level), using joint CLIP, low level, and semantic pipelines.

# D    Performance comparison

**Comparison metrics**    Our study uses various metrics to evaluate how well we can recreate visual stimulus from brain data (EEG, MEG, fMRI) (Tab. 1 and Tab. 7). These metrics include PixCorr (pixelwise correlation, between ground truth and reconstructions), SSIM (structural similarity index metric)[55], SwAV (SwAV-ResNet50, refer to average correlation distance)[56], and two-way identification using neural networks (AlexNet(2/5), Inception, CLIP. Here AlexNet(2/5) the 2nd and 5th feature layers of AlexNet) for both low-level and high-level image features. Here two-way identification can be seem as a two-way retrieval task described in [20]. In Tab. 1, our results showed that on the THINGS dataset, we could achieve performance over MEG on EEG reconstruction using ATM. Tab. 6 shows the decoding performance of different data sets (fMRI, MEG, EEG) on visual

stimulus tasks, and we even achieved the same or better performance than fMRI and MEG. Our results suggest that a suitable neural representation plays a decisive role in the downstream task.

Table 6: The classification performance of various methods are discussed. Due to differences in datasets and data modalities, we have specified unified metrics to objectively assess the performance of each method.

| Dataset | Model | 50-way | | 100-way | | 200-way | |
|---|---|---|---|---|---|---|---|
| | | top-1 | top-5 | top-1 | top-5 | top-1 | top-5 |
| GOD-Wiki (fMRI) | CADA-VAE (V&T)[57] | 10.02 | 40.37 | - | - | - | - |
| | MVAE (V&T) [58] | 10.04 | 39.60 | - | - | - | - |
| | MMVAE (V&T) [59] | 11.68 | 43.29 | - | - | - | - |
| | MoPoE-VAE (V&T) [60] | 12.90 | 51.78 | - | - | - | - |
| | BraVL (V&T) [61] | 13.99 | 53.13 | - | - | - | - |
| THINGS (MEG) | **ATM (Ours)** | 15.63 | 41.38 | 11.75 | 29.25 | 5.88 | 19.25 |
| THINGS (EEG) | BraVL [61] | 14.33 | 40.28 | - | - | 5.82 | 17.45 |
| | **ATM (Ours)** | 17.40 | 39.40 | 11.50 | 28.50 | 7.40 | 20.60 |

Table 7: Quantitative comparison results of image reconstruction in Subject 8 via our framework using different encoders.

| Dataset | Low-level | | High-level | | | | |
|---|---|---|---|---|---|---|---|
| | PixCorr ↑ | SSIM ↑ | AlexNet(2) ↑ | AlexNet(5) ↑ | Inception ↑ | CLIP ↑ | SwAV ↓ |
| NSD-fMRI [12] | 0.305 | 0.366 | 0.962 | 0.977 | 0.910 | 0.917 | 0.410 |
| THINGS-MEG [12] | 0.058 | 0.327 | 0.695 | 0.753 | 0.593 | 0.700 | 0.630 |
| THINGS-MEG (averaged) [12] | 0.090 | 0.336 | 0.736 | 0.826 | 0.671 | 0.767 | 0.584 |
| **THINGS-MEG (Ours)** | **0.104** | **0.340** | **0.613** | **0.672** | **0.619** | **0.603** | **0.651** |
| THINGS-EEG (NICE) [5] | 0.142 | 0.276 | 0.739 | 0.832 | 0.659 | 0.722 | 0.612 |
| THINGS-EEG (EEGNetV4)[31] | 0.140 | 0.302 | 0.767 | 0.840 | 0.713 | 0.773 | 0.581 |
| **THINGS-EEG (Ours)** | **0.160** | **0.345** | **0.776** | **0.866** | **0.734** | **0.786** | **0.582** |

# E   Representational analysis

As depicted in Fig. 12, we showcase the representational similarity matrix and visualization in the latent space. To investigate the relationship between the representations obtained from EEG and those of images, we conducted a representational similarity matrix. We focused on Subject 8, who exhibited the highest retrieval accuracy. By applying a clustering algorithm to the image embeddings corresponding to 200 images in the test set, we observed distinct within-category clustering. We generated similarity matrices based on both image and text embeddings, which were then compared with EEG representations. As shown in Fig. 12, clear within-category clustering is observable in the representational similarity matrix with image, whereas this phenomenon is not present in the representational similarity matrix with text.

# F   Concept analysis

We have adopted the concept embedding encoder proposed by Wei et al. [62], which encodes the clip embedding of the original image into a 42-dimensional vector, with each dimension representing a distinct concept. Utilizing the ATM for direct projection on the EEG data of 200 categories from the test set, we obtained EEG embeddings that were then fed into the concept encoder with frozen weights, yielding 200 concept embeddings, as depicted in Fig. 13. An analysis of representational similarity at the concept level indicates that the EEG embeddings derived from our EEG projector effectively align with the conceptual space. This ensures semantic consistency at a high level of alignment, providing compelling evidence for the reconstruction of images from EEG data.

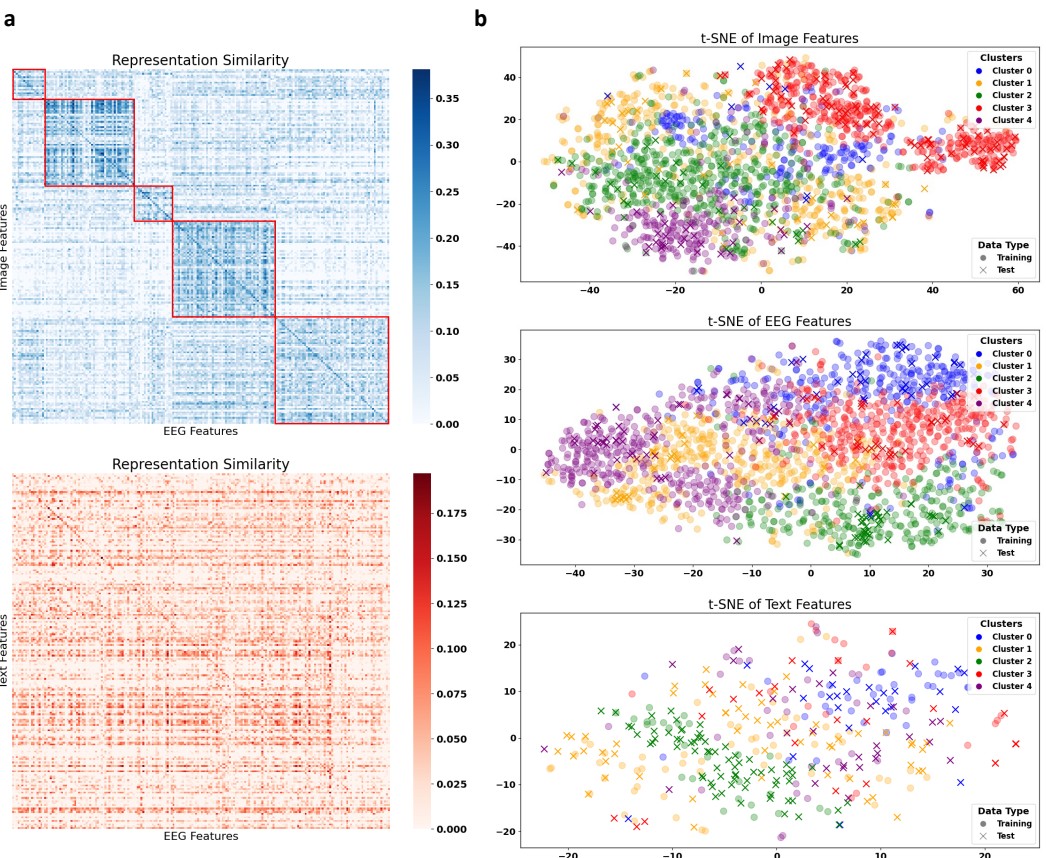

Figure 12: **Visualization of the representation of EEG, image and text modality**. (a) Representational similarity matrix between EEG features and image/text features. (b) Visualization in the latent space of EEG/image/text by t-SNE.

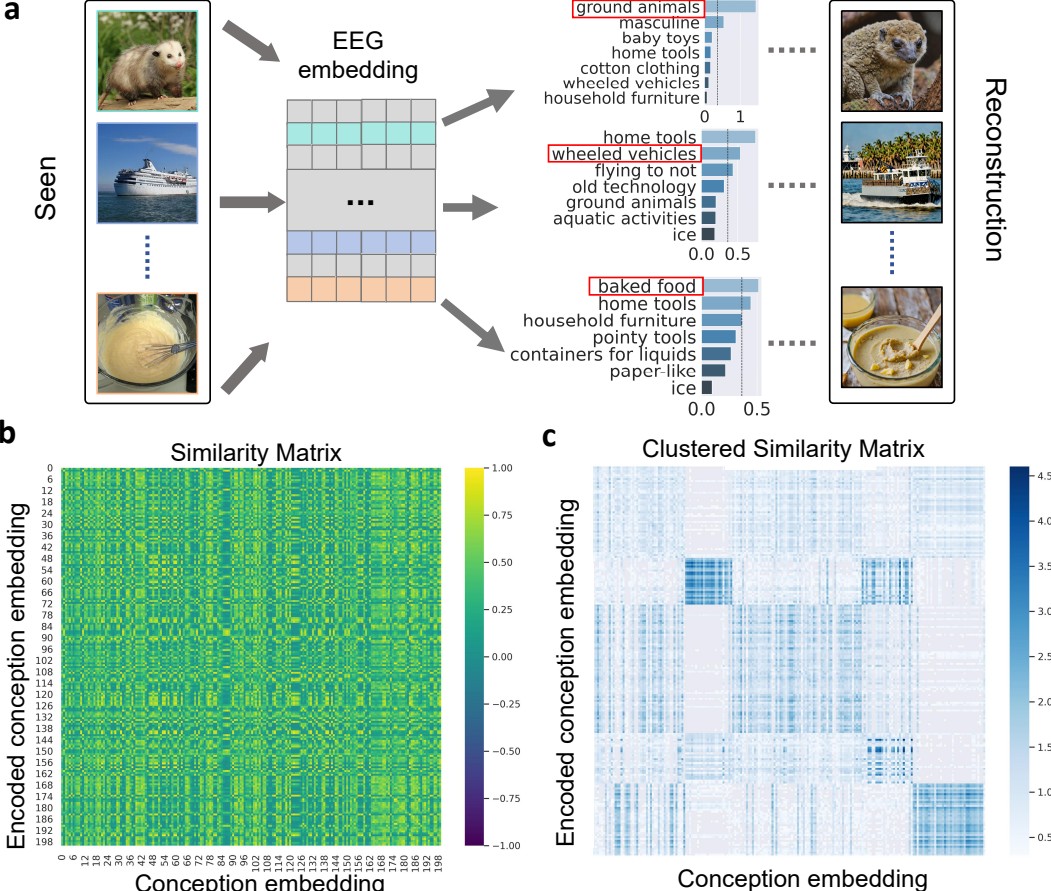

Figure 13: **Visualization of the conceptual representation analysis**. (a) Conceptual representations were obtained from eeg embeddings using concept encoder. (b) The similar matrix between EEG embeddings and real concept embeddings. (c) Concept embedding similarity matrix after cluster rearrangement (k=5).

# G  Additional images results

We visualize the best, medium and worst generated images in Fig. 14. We randomly selected the EEG data of a subject viewing 100 images, and extracted EEG embeddings to guide image generation. By calculating the cosine similarity of the CLIP embedding between the generated image and the original image, we found 12 images each with the best, medium and worst generation effects. It can be seen that in the best group, the generated image is not only highly consistent with the semantics of the original image, but also well retains the low-level visual features. in the medium group, the generated image maintains the semantic features of the original image, and the low-level visual features are well preserved. Visual features were altered. in the worst group, both semantic features and low-level visual features were altered.

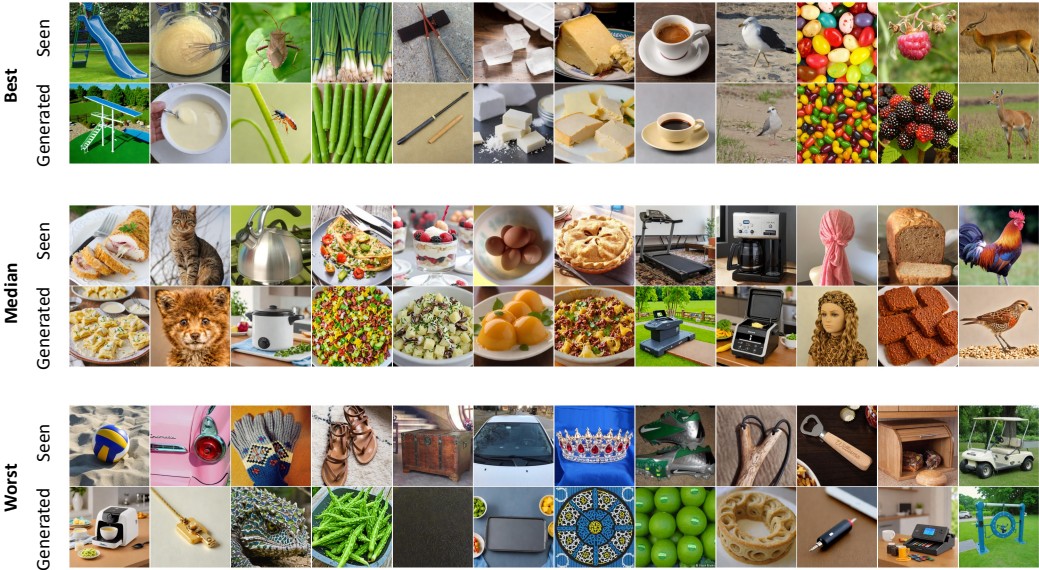

Figure 14: **Examples of EEG-guided visual reconstruction**. From top to bottom, we exhibit the best, median, and worst 12 generated images, respectively. We show the images subjects seen and the generated images by our two-stage image generator.

## G.1  Additional retrieval results

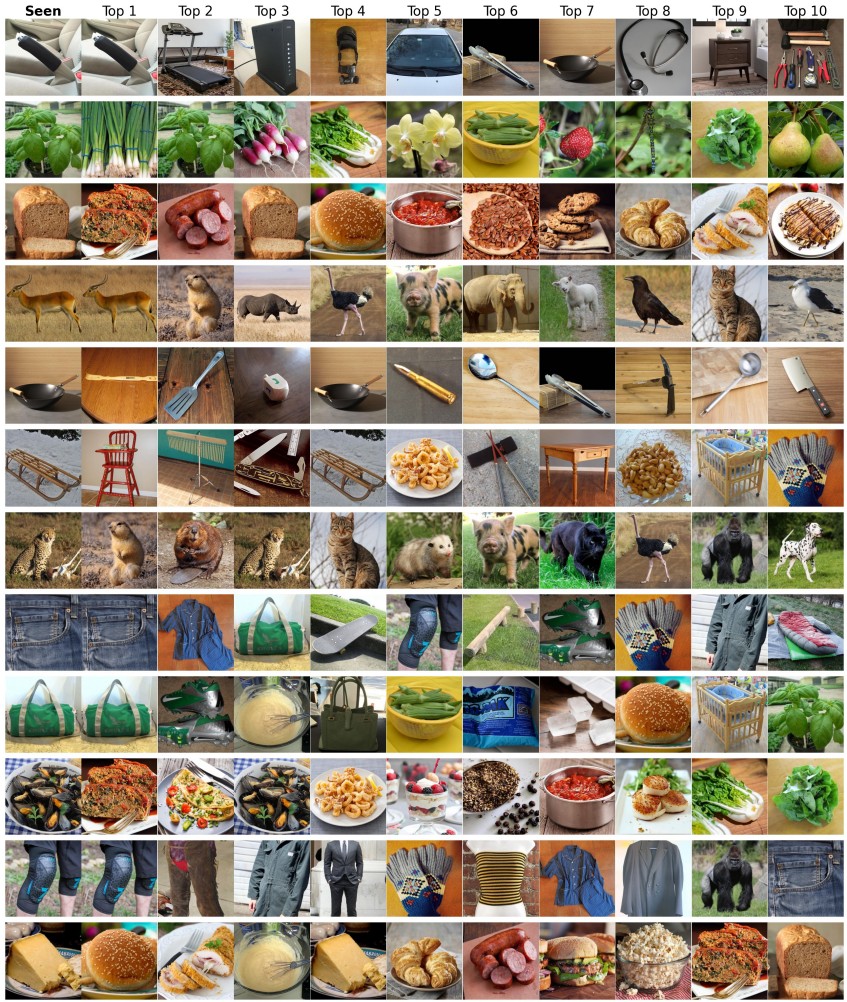

Figure 15: Additional retrieval results

## G.2  Additional generated images

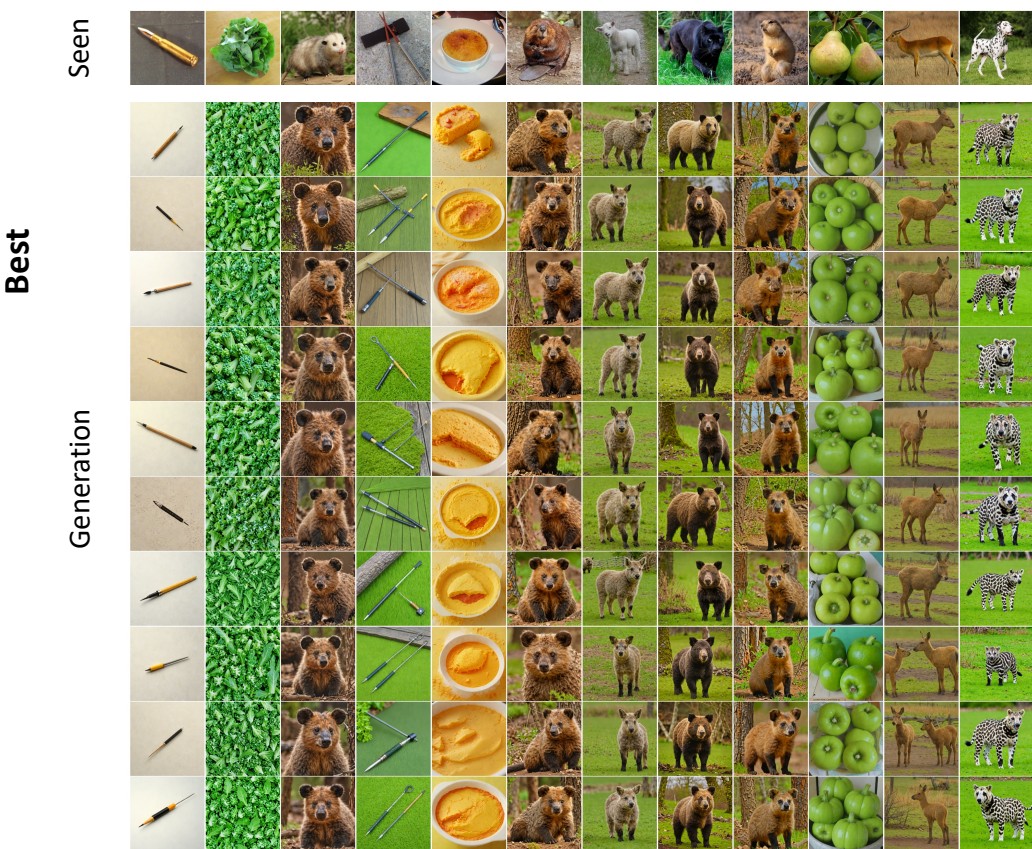

Figure 16: Additional generated results with the best alignment to original images

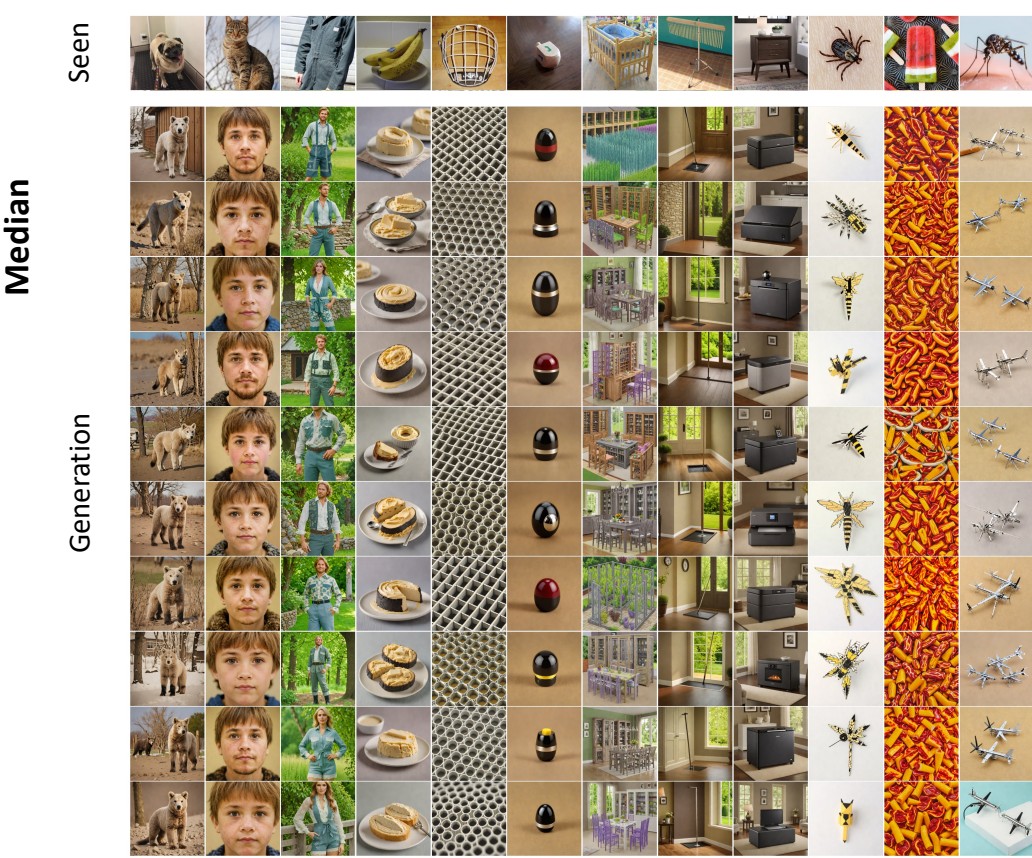

Figure 17: Additional generated results with the median alignment to original images

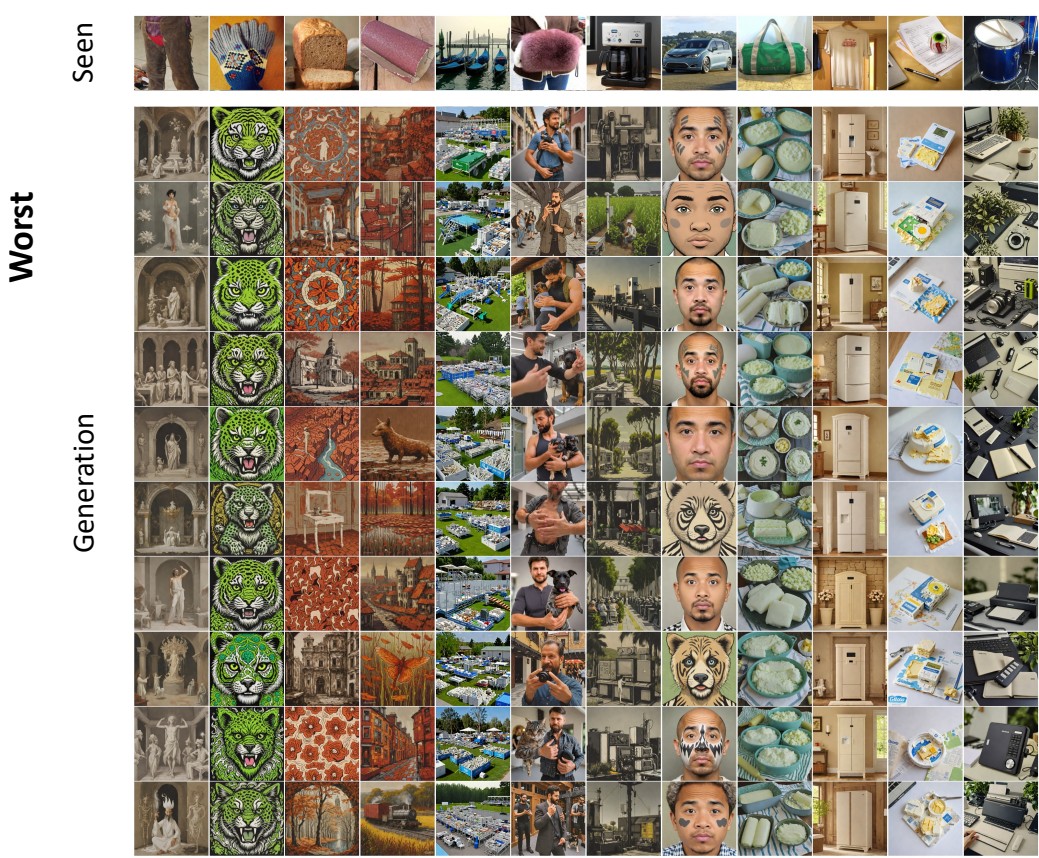

Figure 18: Additional generated results with the worst alignment to original images

## G.3 Additional generated images for each subject

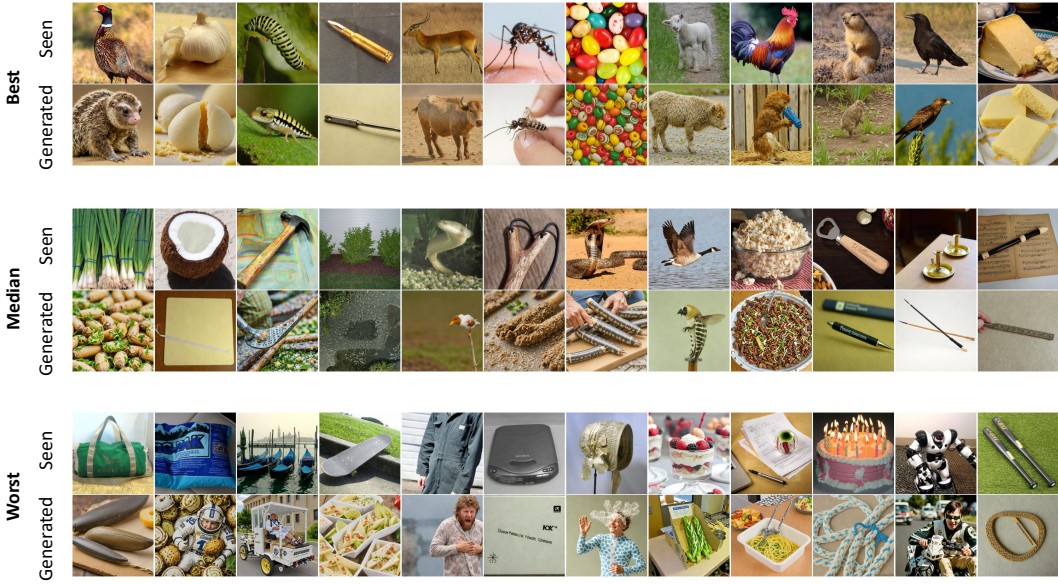

Figure 19: **Part of subject 1 generates images**. We do a batch generation of the subjects and then calculate the best, medium, and worst performers compared to the original stimulus pictures.

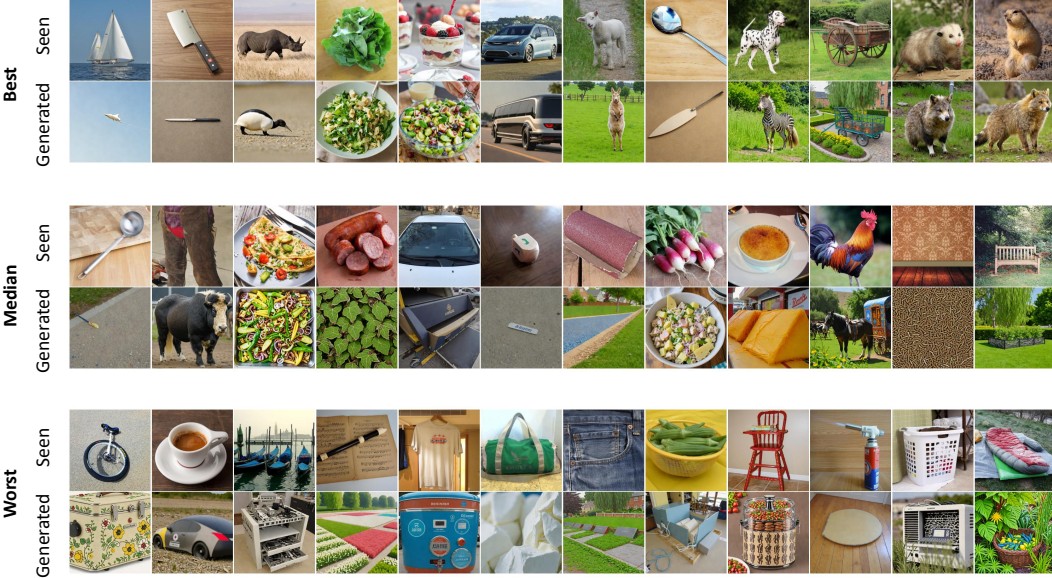

Figure 20: **Part of subject 2 generates images**. We do a batch generation of the subjects and then calculate the best, medium, and worst performers compared to the original stimulus pictures.

## H    Additional evaluation results

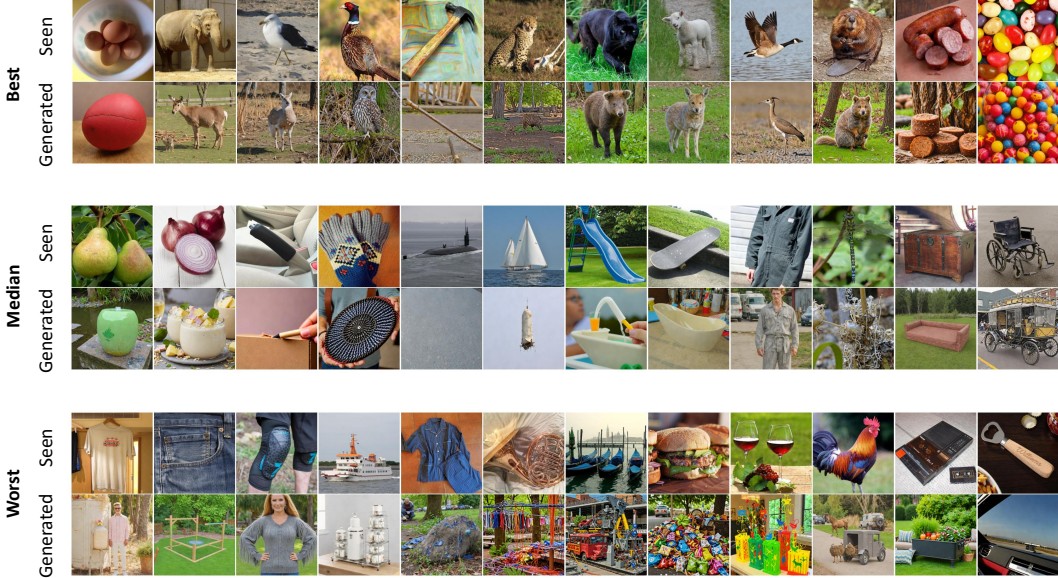

Figure 21: **Part of subject 3 generates images**. We do a batch generation of the subjects and then calculate the best, medium, and worst performers compared to the original stimulus pictures.

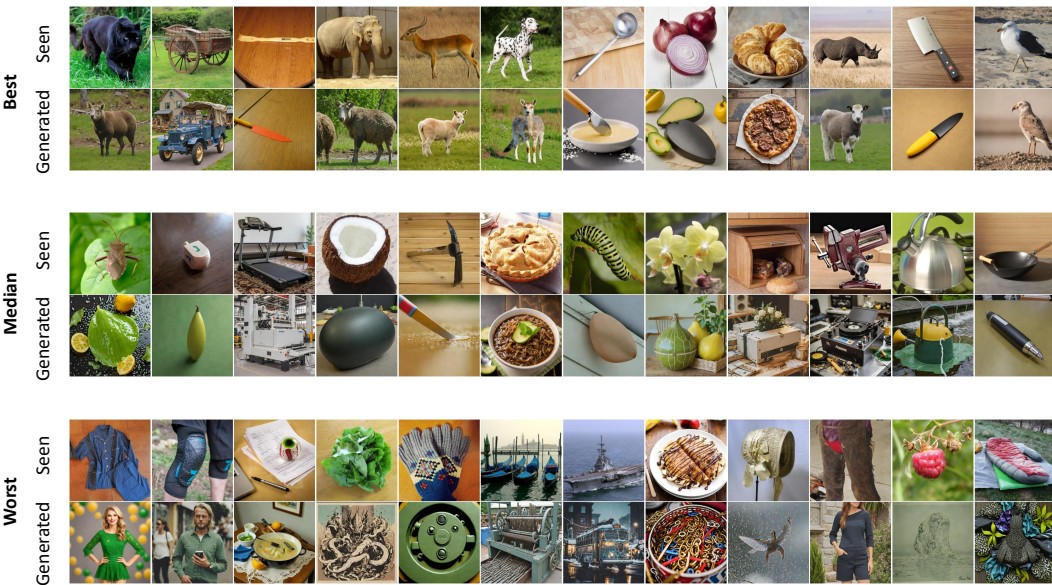

Figure 22: **Part of subject 4 generates images**. We do a batch generation of the subjects and then calculate the best, medium, and worst performers compared to the original stimulus pictures.

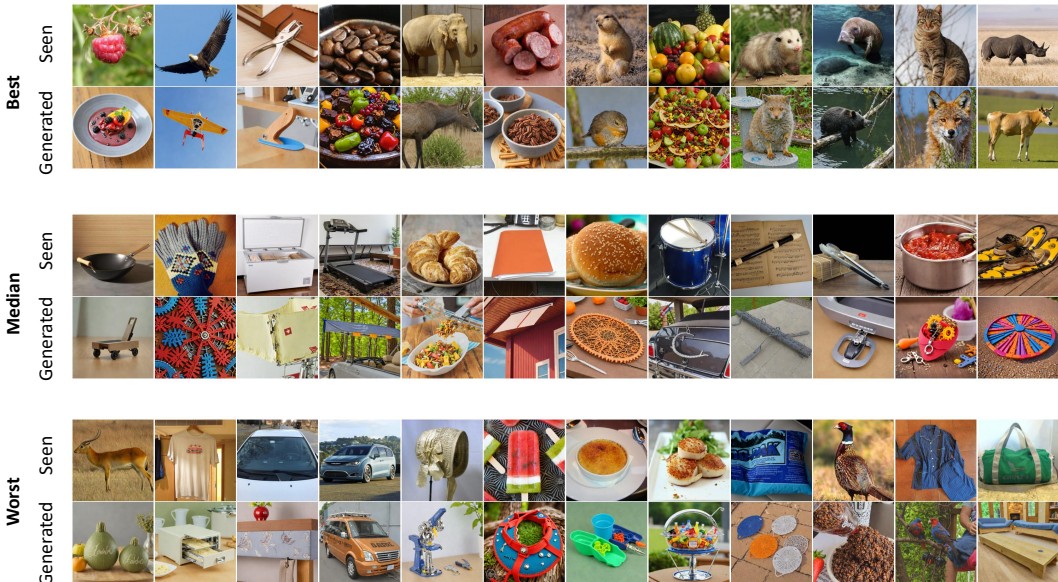

Figure 23: **Part of subject 5 generates images**. We do a batch generation of the subjects and then calculate the best, medium, and worst performers compared to the original stimulus pictures.

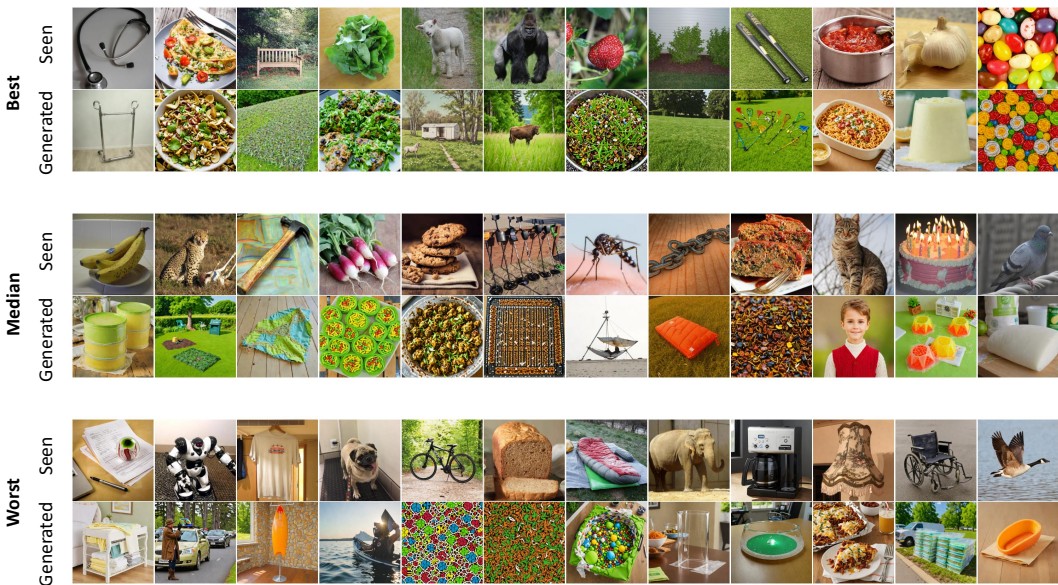

Figure 24: **Part of subject 6 generates images**. We do a batch generation of the subjects and then calculate the best, medium, and worst performers compared to the original stimulus pictures.

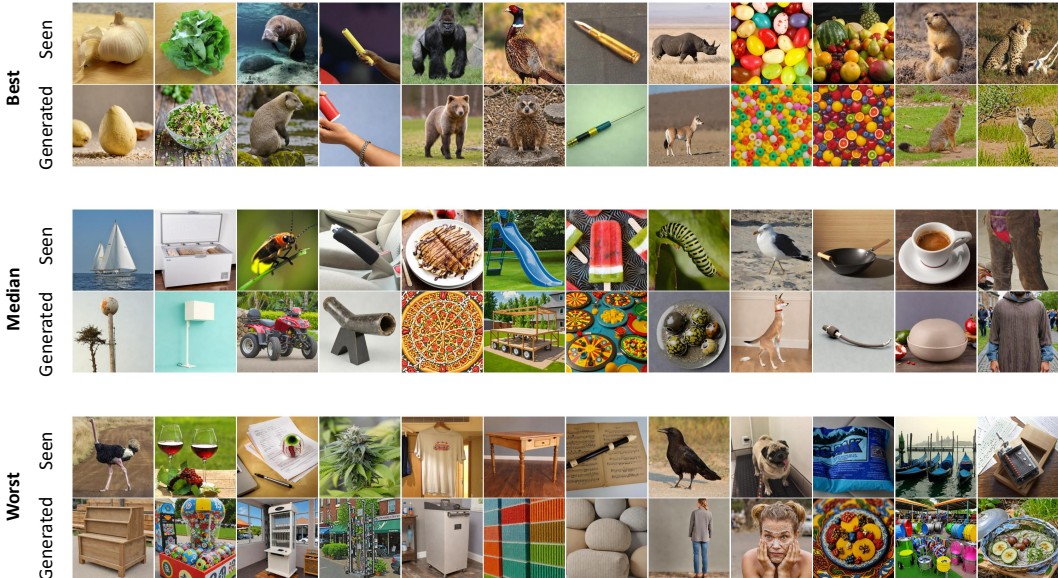

Figure 25: **Part of subject 7 generates images**. We do a batch generation of the subjects and then calculate the best, medium, and worst performers compared to the original stimulus pictures.

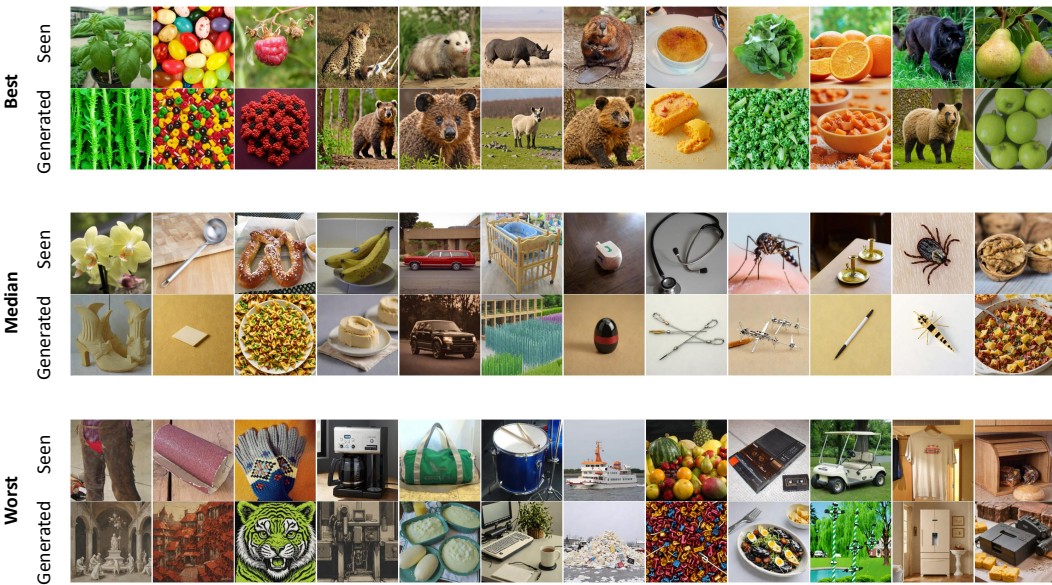

Figure 26: **Part of Subject 8 generates images**. We do a batch generation of the subjects and then calculate the best, medium, and worst performers compared to the original stimulus pictures.

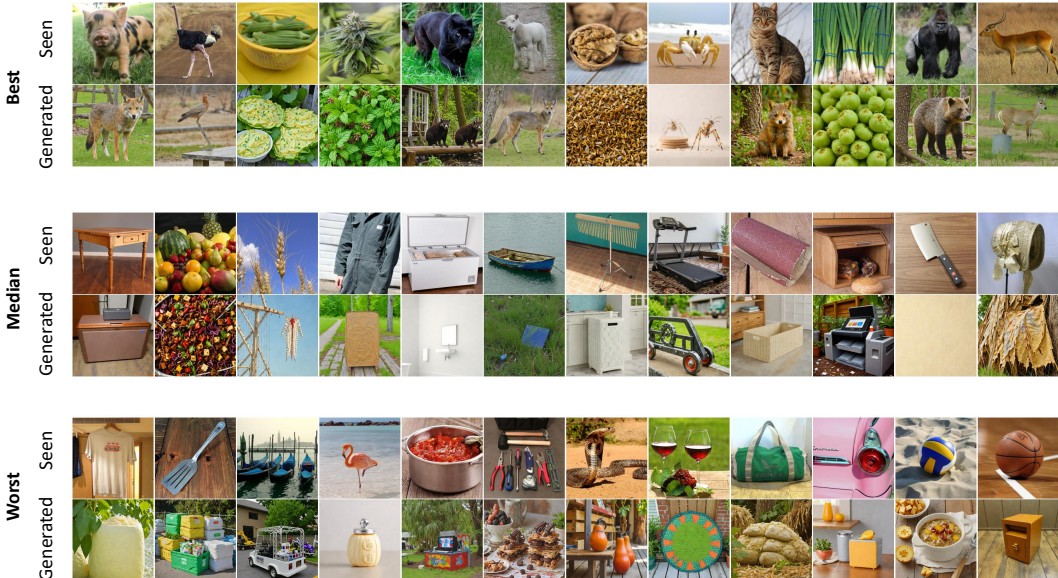

Figure 27: **Part of subject 9 generates images**. We do a batch generation of the subjects and then calculate the best, medium, and worst performers compared to the original stimulus pictures.

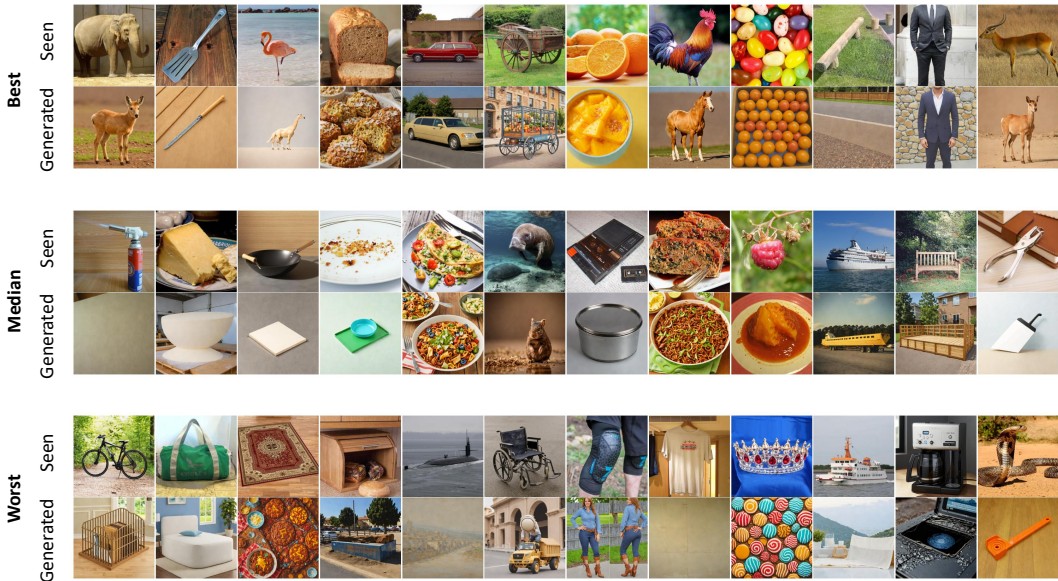

Figure 28: **Part of subject 10 generates images**. We do a batch generation of the subjects and then calculate the best, medium, and worst performers compared to the original stimulus pictures.

## H.1 Accuracy for time windows

According to our results in 30 for time windows, which shows that for all embeddings, a clear peak can be observed for windows ending around 200-250 ms after image onset. Comparing 29 and 30, we can see that, unlike [12], our time window results on THINGS-EEG dataset do not have a second peak, which may be mainly affected by the experimental paradigm. In 7, we can see that in the first 50ms-200ms, the image reconstructed at 50 ms is completely messy and has no semantics; the semantics of the reconstructed image after 250 ms is basically correct and gradually stabilizes, and after 500ms, due to the lack of additional visual response, the content of the image reconstruction is more stable. Similar results are also shown in Figure 4 A of [12]: their method can also reconstruct high-quality images at 100ms. This just shows that our reconstruction results are in line with the neuroscience prior. However, this does not mean that the EEG data after the absence of visual response (200ms) loses its contribution to decoding, because the processing of high-level visual features (corresponding to the visual features of CLIP) may be involved over time.

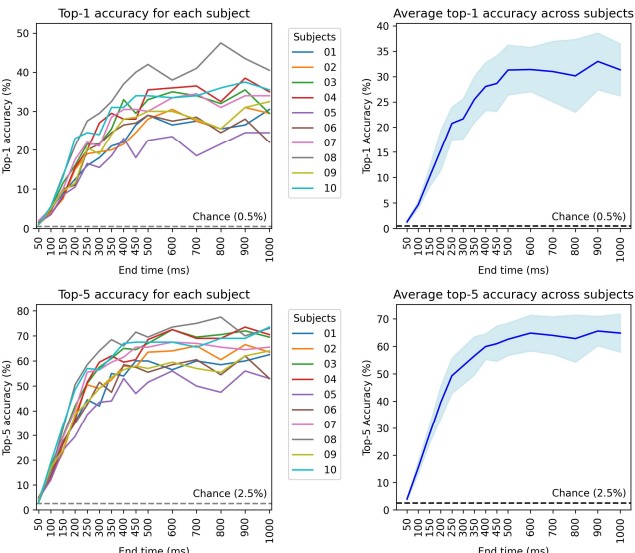

Figure 29: **Accuracy for growing windows**. We use an EEG time window of 100ms, sliding 100ms each time. (a) Top-1 accuracy. (b) Top-5 accuracy.

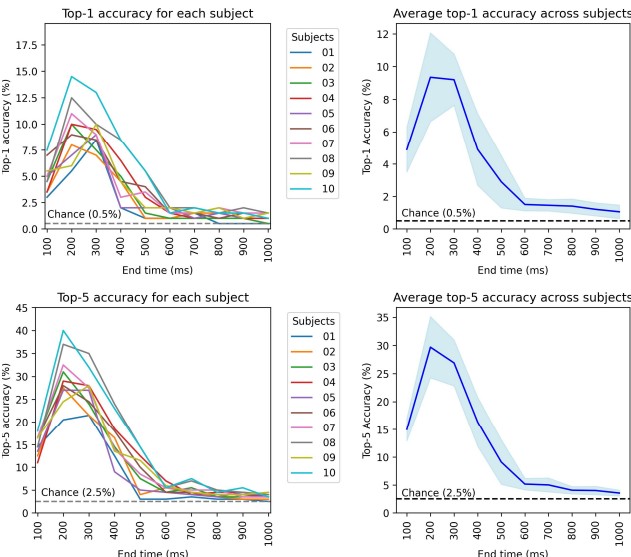

Figure 30: **Accuracy for sliding windows**. We use an EEG time window of 100ms, sliding 100ms each time. (a) Top-1 accuracy. (b) Top-5 accuracy.

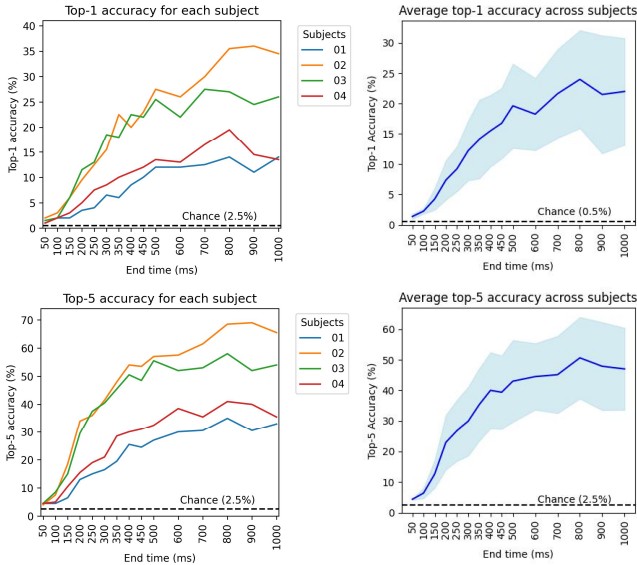

Figure 31: **Accuracy for growing windows**. The MEG time window grows from 50ms to 1000ms. (a) Top-1 accuracy. (b) Top-5 accuracy.

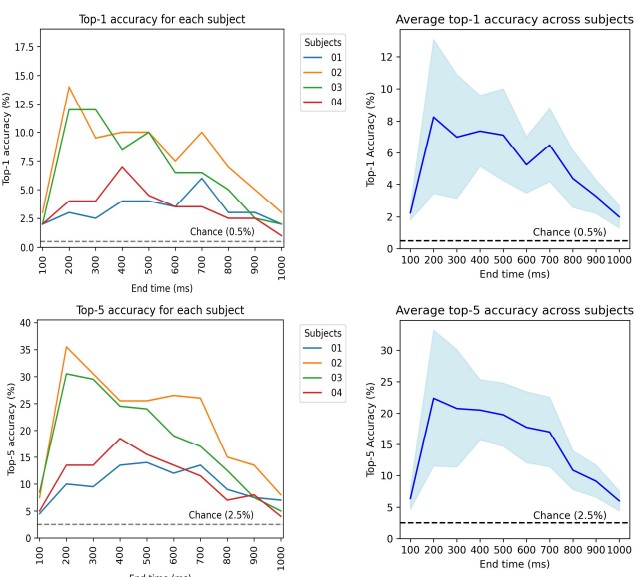

Figure 32: **Accuracy for sliding windows**. We use an MEG time window of 100ms, sliding 100ms each time. (a) Top-1 accuracy. (b) Top-5 accuracy.

Table 8: Overall performance of zero-shot Retrieval on **THINGS-EEG** dataset. We divided the last batch from the original training set as the validation set, selected the best model according to the minimum validation loss in 40 epochs, and finally evaluated the performance on the test set. We showed the performance of in-subject and cross-subject retrieval task (Ave $\pm$ Std.%) under the condition of **batch size=1024**. We compared the 2-way, 4-way, 10-way, the Top-1 and Top-5 accuracy of 200-way from different EEG embedding methods.

| Subject dependent - train and test on one subject (batch size=1024) | | | | | |
|---|---|---|---|---|---|
| **Methods** | **2-Way** | **4-Way** | **10-Way** | **Top-1** | **Top-5** |
| EEGITNet | 76.69 $\pm$ 12.97 | 56.98 $\pm$ 16.31 | 36.35 $\pm$ 15.11 | 5.75 $\pm$ 3.62 | 18.14 $\pm$ 9.40 |
| EEGConformer | 76.17 $\pm$ 13.13 | 56.29 $\pm$ 16.70 | 34.72 $\pm$ 14.79 | 3.98 $\pm$ 2.80 | 17.10 $\pm$ 9.21 |
| ShallowFBCSPNet | 74.32 $\pm$ 12.14 | 53.97 $\pm$ 15.81 | 33.48 $\pm$ 14.35 | 6.10 $\pm$ 4.61 | 16.53 $\pm$ 9.94 |
| EEGNetV4 | 92.81 $\pm$ 2.22 | 83.15 $\pm$ 4.20 | 67.81 $\pm$ 6.11 | 19.51 $\pm$ 5.19 | 48.99 $\pm$ 6.75 |
| B.D. | 78.42 $\pm$ 8.81 | 58.24 $\pm$ 12.13 | 37.97 $\pm$ 11.38 | 5.88 $\pm$ 3.49 | 18.61 $\pm$ 7.81 |
| NICE | 93.69 $\pm$ 2.15 | 84.27 $\pm$ 5.27 | 69.91 $\pm$ 8.21 | 21.52 $\pm$ 5.90 | 51.57 $\pm$ 10.97 |
| MLP | 84.08 $\pm$ 3.42 | 67.39 $\pm$ 5.71 | 46.29 $\pm$ 6.23 | 7.34 $\pm$ 2.14 | 24.39 $\pm$ 5.18 |
| **ATM-S (Ours)** | **94.92 $\pm$ 1.45** | **87.91 $\pm$ 3.14** | **75.37 $\pm$ 5.77** | **26.13 $\pm$ 8.15** | **55.32 $\pm$ 10.57** |
| **ATM-E (Ours)** | 92.99 $\pm$ 2.20 | 83.81 $\pm$ 4.46 | 68.87 $\pm$ 7.27 | 22.40 $\pm$ 6.62 | 50.59 $\pm$ 9.59 |
| Subject independent - leave one subject for test (batch size=1024) | | | | | |
| **Methods** | **2-Way** | **4-Way** | **10-Way** | **Top-1** | **Top-5** |
| EEGNetV4 | 82.85 $\pm$ 3.62 | 64.65 $\pm$ 6.29 | 42.35 $\pm$ 7.10 | 6.25 $\pm$ 2.56 | 20.95 $\pm$ 5.73 |
| EEGConformer | 56.54 $\pm$ 4.00 | 31.80 $\pm$ 3.20 | 13.89 $\pm$ 2.01 | 0.87 $\pm$ 0.33 | 4.42 $\pm$ 1.20 |
| ShallowFBCSPNet | 75.76 $\pm$ 2.49 | 53.63 $\pm$ 3.58 | 31.43 $\pm$ 4.30 | 2.51 $\pm$ 1.31 | 12.03 $\pm$ 2.78 |
| EEGNetV4 | 82.60 $\pm$ 3.17 | 64.28 $\pm$ 5.44 | 42.24 $\pm$ 6.10 | 6.13 $\pm$ 2.40 | 21.23 $\pm$ 5.19 |
| B.D. | 72.84 $\pm$ 12.41 | 51.41 $\pm$ 15.10 | 31.67 $\pm$ 12.96 | 4.10 $\pm$ 2.72 | 14.46 $\pm$ 7.97 |
| NICE | 83.88 $\pm$ 2.57 | 66.14 $\pm$ 5.30 | 45.13 $\pm$ 5.85 | 8.24 $\pm$ 3.01 | 23.76 $\pm$ 5.09 |
| MLP | 75.80 $\pm$ 2.45 | 55.08 $\pm$ 3.07 | 34.05 $\pm$ 2.83 | 4.46 $\pm$ 0.81 | 15.26 $\pm$ 2.34 |
| **ATM-S (Ours)** | **83.72 $\pm$ 3.01** | **66.91 $\pm$ 5.41** | **46.53 $\pm$ 6.24** | **8.24 $\pm$ 2.73** | **25.36 $\pm$ 6.24** |
| **ATM-E (Ours)** | 80.65 $\pm$ 3.11 | 61.65 $\pm$ 5.31 | 39.66 $\pm$ 6.44 | 7.00 $\pm$ 2.20 | 21.12 $\pm$ 5.25 |

