# OpenReview forum: "Visual Decoding and Reconstruction via EEG Embeddings with Guided Diffusion"
_NeurIPS.cc/2024/Conference — NeurIPS 2024 poster_

### Official Review · Reviewer_gPm4 · 2024-07-09

**Soundness:** 2
**Presentation:** 3
**Contribution:** 3
**Rating:** 5
**Confidence:** 4

**Summary:**

This paper presents an end-to-end EEG-based visual decoding framework that includes two stages: a brain encoder for EEG feature extraction and a "generator" for producing reconstructed images. The experiments demonstrate effective results in retrieval, classification, and reconstruction tasks across two datasets, suggesting potential applications for real-time brain-computer interfaces.

**Strengths:**

The research demonstrates the feasibility of using non-invasive EEG signals for image decoding as using fMRI techniques. The analysis provides a valuable reference for EEG feature extraction.

The paper highlights three primary contributions: (1) an EEG-image decoding framework, (2) a novel EEG encoder, and (3) a two-stage EEG-to-image generation strategy that leverages both low- and high-level visual information.

**Weaknesses:**

The paper's novelty in the context of brain-image decoding methodologies is not distinctly clear.

1. The pipeline, which includes a contrastive learning-driven encoder and a diffusion-based decoder, is already well-established in the field.

2. The ATM encoder, which incorporates channel-wise attention, spatial convolution, and temporal convolution, does not significantly differ from previous studies, such as mentioned in the introduction, Benchetrit et al. and Song et al.

3. The method for disentangling low- and high-level features within the framework is unclear, particularly how the VAE is supposed to provide low-level image features.

The evaluation section contains many conclusions that conflict with scientific validity, which may cause serious misleading.

1. How can we get clear reconstruction results with only 50 ms signals after the onset as in Figure 7c? The visual stimuli haven’t reached V1 in such a short time.

2. We know the temporal cortex is strongly related to object recognition. But in Figure 8b, the temporal channels contribute very small. Based on that, should we think the retrieval task was based on some low-level information?

3. The visual response is very quick after the onset, finished before 200 ms. It cannot be thought of as visual responses at 500 ms after the onset on Page 7 Line 182, and contained within 200-400 ms on Page 9 Line 246.

4. On page 9 Line 251, It can’t be concluded that EEG is better than MEG in visual tasks where the paradigms used for these two datasets were different.

**Questions:**

1. The paper mentions selecting the highest test accuracy during the training process as the statistical result. It would be more rigorous to test the model only once after it is fully trained.

2. In the framework illustrated in Figure 2, which input to the diffusion process is most critical for image generation—the reconstructed output after VAE, the image embedding transferred from EEG, or the generated caption?

3. How does the model ensure that it captures low-level features after the VAE encoder?

4. What is the impact of the pre-trained language and generative models on the final performance?

5. Could you provide more details about the cross-subject settings described in P3L91? Specifically, what roles do the subject token and shared token in Figure 3 play, and do they enhance cross-subject capability?

6. Figure 11a seems to show no significant correlation between text and EEG features. Could you comment on this observation?

**Limitations:**

Yes. The authors adequately addressed the limitations and future works.

---

> ### Author Rebuttal · Authors · 2024-08-07
>
> Thank you for your careful review and comments. Below please find our point-to-point responses to your comments.
>
> **Q1. “The pipeline is already well-established in the field.”**
>
> Please kindly see the relevant answer in Global rebuttal: Q1 and Q2.
>
> **Q2. How the VAE is supposed to provide low-level image features.**
>
> Thanks to the reviewer for pointing out this part, and we may not have expressed it clearly in the manuscript. Due to limited space, please refer to the reply to **reviewer (zVcP)**.
>
> **Q3. “How can we get clear reconstruction results when only 50 ms? ”**
>
> Please kindly see the relevant answer in Global rebuttal: Q3.
>
> **Q4. Should we think the retrieval was based on some low-level information?**
>
> We consider the high-level visual features should play a major role in the visual retrieval task.
>
> On the one hand, there is much noise in the scalp EEG, and the dynamic activities of the temporal cotex may not be accurately captured by the EEG, while the occipital cortex continues to have a strong response.
>
> Furthermore, when we perform the retrieval task, we use the high-level visual features after the alignment of CLIP and EEG. So the semantic-level visual features should play a major role in the visual retrieval task.
>
> **Q5. The visual response is very quick after the onset, finished before 200 ms....**
>
> Please also kindly see the relevant answer in Global rebuttal: Q3.
>
> **Q6. On page 9 Line 251, It can’t be concluded that...**
>
> In THINGS-MEG, each image in the THINGS-MEG was displayed for 500 ms. There was a fixed time for each image of 1000 ± 200 ms.
>
> However, in THINGS-EEG, each of image was presented for 100 ms. For preprosessing and training, we segmented the EEG data from 0 to 1000 ms after the stimulus onset into trials. We speculate that our excellent performance in the EEG visual decoding task is due to the larger data size of the THINGS-EEG (the more trials, the better the decoding performance). That is, in the visual decoding task, scaling low still works.
>
> Due to the different experimental paradigms of THINGS-EEG and THINGS-MEG and the characteristics of the two types of data, we only report the phenomena we observed. We hope that these results can trigger further discussion and promote the development of the community.
>
>
> **Q7. It would be more rigorous to test the model only once after it is fully trained.**
>
> As described in the manuscript, what we used in Figure 5c and Figure 5d is the result of the highest test accuracy of all methods in the process of training. In Table 7 in the Appendix G, we present the results after 10 tests and take the average results. Regardless of the test strategy, our method performs far ahead of other methods.
>
> **Q8 .Which input to the diffusion process is most critical for image generation ?**
>
> This is also an interesting question. In fact, after our tests, the EEG embedding aligned with CLIP is the most necessary. The output after VAE reconstruction also comes from EEG and the reconstructive image is very blurry, so it contributes very little. On the other hand, the text obtained from the generated caption is very similar to the image embedding, so it also contributes very little.
>
> **Q9. How does the model ensure that it captures low-level features?**
>
> Please kindly find the related explanations in reviewer (zVcP) in Q2. The low-level and high-level here refer to the representation that obtained from the pixel-level or the semantic-level after CLIP alignment with [1][2]. So it is not the low-level and high-level areas of visual cotex in neuroscience.
>
> **Q10. What is the impact of the pre-trained models on the final performance?**
>
> During the experiment, we found that the pre-trained language model seemed to have little impact on the final results, whether in the representation learning stage or the generative model stage. This may be because CLIP itself is a pre-trained model after the image and text are aligned. So the contribution of the text is not so important.
>
> On the other hand, the generative model only reconstructs the semantically correct category. Therefore, it does not improve the decoding accuracy, but only guarantees the quality of the reconstructed image.
>
> **Q11. Provide more details about the cross-subject settings, and do they enhance cross-subject capability?**
>
> Please find the explanations in reviewer (zVcP) in Q7. The objective is to retain both subject-independent joint EEG representations and subject-specific independent EEG representations during training.
>
> **Q12. Could you comment on this observation in Figure 11a ?**
>
> As explained in Q10, this may be due to the fact that CLIP is a pre-trained model with aligned image and text features. We consider that these works [3][4] that use CLIP to align neural signals will also have this property.
>
> **Reference**
>
> [1]Scotti P, Banerjee A, Goode J, et al. Reconstructing the mind's eye: fMRI-to-image with contrastive learning and diffusion priors[J]. Advances in Neural Information Processing Systems, 2024, 36.
>
> [2]Takagi Y, Nishimoto S. High-resolution image reconstruction with latent diffusion models from human brain activity[C]//Proceedings of the IEEE/CVF Conference on Computer Vision and Pattern Recognition. 2023: 14453-14463
>
> [3]Du C, Fu K, Li J, et al. Decoding visual neural representations by multimodal learning of brain-visual-linguistic features[J]. IEEE Transactions on Pattern Analysis and Machine Intelligence, 2023, 45(9): 10760-10777.
>
> [4]Zhou Q, Du C, Wang S, et al. CLIP-MUSED: CLIP-Guided Multi-Subject Visual Neural Information Semantic Decoding[C]//The Twelfth International Conference on Learning Representations.
>
> [5]Benchetrit Y, Banville H, King J R. Brain decoding: toward real-time reconstruction of visual perception[C]//The Twelfth International Conference on Learning Representations.
>
> [6]Song Y, Liu B, Li X, et al. Decoding Natural Images from EEG for Object Recognition[C]//The Twelfth International Conference on Learning Representations.

---

> ### Comment · Reviewer_gPm4 · 2024-08-13
> **Response to authors**
>
> Thanks for your kind reply. I still have concerns: 1) Q7, Why not use a validation set? It's not convincing to use test sets in training processing. 2) Q9, what are the gains of low-level features from VAE for the overall framework, if they are pixel-level representations after CLIP alignment, instead of low-level features defined in vision?

---

> ### Author Response · Authors · 2024-08-13
> **Response to Reviewer gPm4**
>
> We appreciate the reviewer's thoughtful feedback and recognize the significance of the highlighted concerns.
>
> (1) Thank you for your suggestion. We will add the results on the validation set in the official version of the paper. In fact, we initially considered that if we adopt the same approach as in [1], that is, to divide a small number of trials in the training set as the validation set for evaluating the model, this may violate the zero-shot task setting. Different from those classification and retrieval tasks with known categories, due to our unique processing form (in a dataset with a total of 1864 categories, we use 1654 categories of samples as the training set and the remaining 200 categories as the test set), so we hope to always evaluate the performance of the model in a zero-shot form.
>
> Our original approach was to train the models of different encoders for a sufficient number of rounds (30 epochs in the experiment) to ensure model convergence, and test them on the test set in the last 10 epochs. Since all models use the same evaluation method, and we strictly control the random seed and ensure that the data is not leaked, the final evaluation results are also unbiased.
>
> (2) This is another interesting question. First of all, our statement has some typo. What we want to express is "The low-level and high-level here refer to the representation that obtained from the pixel-level after VAE alignment with or the semantic-level after CLIP alignment with".
> Past work [2] has found that in the denoising stage of the early diffusion model, z signals (corresponding to the VAE latent in our framework) dominated prediction of fMRI signals. And during the middle step of the denoising process, zc predicted activity within higher visual cortex much better than z. However, please note that this is only an analysis based on decoding accuracy. These analyses do not have a strong neuroscience causal relationship. We still cannot conclude that the low-level features of fMRI are modeled by VAE.
>
> From our experimental results, the more VAE latent is used in the denoising process, the more certain the overall reconstructed image is and the less details it has. Conversely, the more details it has. Therefore, the contribution of VAE latent and clip latent to reconstruction tends to be a balance relationship. Our future work should focus on achieving more brain-like decoding, paying attention to both well low-level and high-level reconstruction, rather than maintaining an either-or relationship between the two.
>
>
> **Reference**
>
> [1] Song Y, Liu B, Li X, et al. Decoding Natural Images from EEG for Object Recognition[C]//The Twelfth International Conference on Learning Representations.
>
> [2] Takagi Y, Nishimoto S. High-resolution image reconstruction with latent diffusion models from human brain activity[C]//Proceedings of the IEEE/CVF Conference on Computer Vision and Pattern Recognition. 2023: 14453-14463

---

> > ### Author Response · Authors · 2024-08-14
> > **Thank you for your further suggestive comments**
> >
> > We further explain the question (1) you have raised. We added the performance of different EEG encoders on the test set with batch sizes of 16 and 1024 in the anonymous code (https://anonymous.4open.science/r/Visual_Reconstruction-AC56). It can be seen that all methods gradually converge with the increase of training epochs and present a performance value with very small variance on the test set.
> >
> > Therefore, **Figure 5c** and **Figure 5d** of this article use the epoch with the highest test set accuracy in the same total epochs, under the statistics of multiple random seeds. The results in **Table 7** are the average test set accuracy of the last 10 epochs after training convergence. In the **one-page pdf** file we uploaded, there is a more comprehensive performance of each subject's data for reference.

---

> ### Author Response · Authors · 2024-08-14
> **Response to Reviewer gPm4**
>
> Sorry to bother you. We are about to run out of time to respond.
>
> We have made complements and explanations for this work with the help of all the reviews. We would be grateful if you could confirm whether the rebuttal meets your expectations and if there is any other suggestion.
>
> Thank you once again for your time and insightful comments!

---

### Official Review · Reviewer_4xL4 · 2024-07-10

**Soundness:** 1
**Presentation:** 2
**Contribution:** 2
**Rating:** 3
**Confidence:** 4

**Summary:**

The paper presents a end-to-end EEG-based visual reconstruction zero-shot framework, featuring the Adaptive Thinking Mapper (ATM) and a two-stage EEG-to-image generation strategy. This method achievies state-of-the-art performance in classification, retrieval, and reconstruction tasks, and significantly advancing the field of brain-computer interfaces.

**Strengths:**

1.Comprehensive EEG experiments, encompassing retrieval, classification, and visual stimulus reconstruction.
2.Cross-subject considerations.

**Weaknesses:**

1.The manuscript has several significant deficiencies. First, its motivation is based on the signal differences between EEG and fMRI, concluding that EEG's performance limitations are due to constraints in decoding and reconstruction frameworks. However, the proposed EEG encoder merely adds a Channel Attention layer compared to NICE [1], with no detailed explanation provided. Additionally, the loss function in Section 2.4 is taken directly from [2], demonstrating a lack of originality. Furthermore, the visual reconstruction framework shows no substantial difference from existing fMRI methods [3,4], as it also utilizes pre-trained stable diffusion models and their variants, with minor differences, such as the incorporation of Sdedit [5], being tricks to enhance generation quality. Thus, the manuscript fails to substantiate its claims and contributions convincingly.
2.The processing of visual information in the brain involves multiple stages and typically takes around 100-150 milliseconds to reach higher visual areas where complex processing occurs. EEG signals at 50 milliseconds are likely still within the retina and optic nerve stages. Thus, the generation of images from 50ms EEG signals, as shown in Figure 7, contradicts established neuroscience principles. This suggests that the visual stimulus reconstruction framework heavily relies on the image generation model, which may have limited significance for the field of neuroscience.

[1] Song, Yonghao, et al. "Decoding Natural Images from EEG for Object Recognition." arXiv preprint arXiv:2308.13234 (2023).
[2] Benchetrit, Yohann, Hubert Banville, and Jean-Rémi King. "Brain decoding: toward real-time reconstruction of visual perception." arXiv preprint arXiv:2310.19812 (2023).
[3] Chen, Zijiao, et al. "Seeing beyond the brain: Conditional diffusion model with sparse masked modeling for vision decoding." Proceedings of the IEEE/CVF Conference on Computer Vision and Pattern Recognition. 2023.
[4] Lu, Yizhuo, et al. "Minddiffuser: Controlled image reconstruction from human brain activity with semantic and structural diffusion." Proceedings of the 31st ACM International Conference on Multimedia. 2023.
[5] Meng, Chenlin, et al. "Sdedit: Guided image synthesis and editing with stochastic differential equations." arXiv preprint arXiv:2108.01073 (2021).

**Questions:**

I have no quentions, please see the weaknesses.

**Limitations:**

Yes

---

> ### Author Rebuttal · Authors · 2024-08-07
>
> Thank you for your careful review and comments. Below please find our point-to-point responses to your comments.
>
> **Q1. “First, its motivation is based on the signal differences between EEG and fMRI, concluding that EEG's performance limitations are due to constraints in decoding and reconstruction frameworks......”**
>
> Please kindly see the relevant answer in **Global rebuttal: Q1 and Q2**. This paper has two main motivations:
>
> First, existing fMRI-based visual decoding and image reconstruction methods [1][2] are very advanced, but they are limited by the temporal resolution, portability and cost of fMRI, so these technologies cannot be applied in practice.
>
> Second, the idea of the NICE [3] and B.D. [2]’s method are simple but very general. Previous work on visual decoding using fMRI often used ridge regression as the encoder. However, using EEG is different from fMRI, as it is full of noise and the original signal-to-noise ratio is low. It requires a sufficiently strong prior and a more complex model structure to extract the feature maps. We carefully studied the shortcomings of different approaches in EEG decoding [5][6]. Inspired by NICE [3], and based on the theory of treating channels as patches in [4], we introduced joint subject training, which far aheads of NICE, and gained the ability to adapt to new subjects like [7][8].
>
> The contribution of this paper is not to improve decoding performance too much only by stacking tricks. The main purpose is to provide evidence for the neuroscience and machine learning communities: using our framework, even compared with fMRI, EEG can provide competitive performance.
>
> **Q2. “The generation of images from 50ms EEG signals, as shown in Figure 7, contradicts established neuroscience principles. This suggests that the visual stimulus reconstruction framework heavily relies on the image generation model, which may have limited significance for the field of neuroscience.”**
>
> Please kindly see the relevant answer in **Global rebuttal: Q3**. We may not have explained it clearly in the article, but there are similar expressions in [2]. According to our results in **Appendix H.1 Accuracy for time windows**, **Figure 29** shows that for all embeddings, a clear peak can be observed for windows ending around 200-250 ms after image onset. Comparing **Figure 28** and **Figure 29**, we can see that, unlike [2], our time window results on THINGS-EEG dataset do not have a second peak, which may be mainly affected by the experimental paradigm.
>
> Please pay attention to the caption in Figure 7 and the reconstruction results from 0 to 50 ms and 0 to 250 ms. It can be seen that the image reconstructed at 50 ms is completely messy and has no semantics; the semantics of the reconstructed image after 250 ms is basically correct and gradually stabilizes. Similar results are also shown in **Figure 4 A of [2]**: their method can also reconstruct high-quality images at 100ms. This just shows that our reconstruction results are in line with the neuroscience prior.
>
> In addition, we provide three different tasks including image classification, image retrieval and image reconstruction. The excellent performance in image classification and image retrieval tasks shows that the decoding of visual stimuli is reasonable. Since the submission is to NeurIPS, the top machine learning conference, it is necessary for us to use the latest technology to ensure the quality of image reconstruction on this basis, which reflects the latest technological progress.
>
> **Reference**
>
> [1]Scotti P S, Tripathy M, Torrico C, et al. MindEye2: Shared-Subject Models Enable fMRI-To-Image With 1 Hour of Data[C]//Forty-first International Conference on Machine Learning.
>
> [2]Benchetrit Y, Banville H, King J R. Brain decoding: toward real-time reconstruction of visual perception[C]//The Twelfth International Conference on Learning Representations.
>
> [3]Song Y, Liu B, Li X, et al. Decoding Natural Images from EEG for Object Recognition[C]//The Twelfth International Conference on Learning Representations.
>
> [4]Liu Y, Hu T, Zhang H, et al. iTransformer: Inverted Transformers Are Effective for Time Series Forecasting[C]//The Twelfth International Conference on Learning Representations.
>
> [5]Lawhern V J, Solon A J, Waytowich N R, et al. EEGNet: a compact convolutional neural network for EEG-based brain–computer interfaces[J]. Journal of neural engineering, 2018, 15(5): 056013.
>
> [6]Song Y, Zheng Q, Liu B, et al. EEG conformer: Convolutional transformer for EEG decoding and visualization[J]. IEEE Transactions on Neural Systems and Rehabilitation Engineering, 2022, 31: 710-719.
>
> [7]Xia W, de Charette R, Öztireli C, et al. UMBRAE: Unified Multimodal Decoding of Brain Signals[J]. arXiv e-prints, 2024: arXiv: 2404.07202.
>
> [8]Zhou Q, Du C, Wang S, et al. CLIP-MUSED: CLIP-Guided Multi-Subject Visual Neural Information Semantic Decoding[C]//The Twelfth International Conference on Learning Representations.

---

> > ### Author Response · Authors · 2024-08-14
> > **Response to Reviewer 4xL4**
> >
> > Thank you for your comments and feedback!
> >
> > **Q1:  ”Its motivation is based on the signal differences between EEG and fMRI, concluding that EEG's performance limitations are due to constraints in decoding and reconstruction frameworks. ”**
> >
> > Although fMRI plays an important role in neuroimaging research, fMRI has low temporal resolution, bulky and non-portable equipment, high cost, and is non-invasive but limited by the magnetic field environment. So fMRI-based work is almost impossible to apply in practice, and this have hindered the development of the brain-computer interface (BCI). This motivates us to propose a zero-shot visual decoding and reconstruction framework that can be proven effective on Image-EEG datasets. Our manuscript provides an empirical guidance on practical BCI applications. We hope that the BCI and neuroscience communities pay more attention to the implementation of similar technologies and real data rather than overfitting only on existing fMRI datasets.
> >
> > **Q2: “However, the proposed EEG encoder merely adds a Channel Attention layer compared to NICE [1], with no detailed explanation provided. Additionally, the loss function in Section 2.4 is taken directly from [2], demonstrating a lack of originality. Furthermore, the visual reconstruction framework shows no substantial difference from existing fMRI methods [3,4], as it also utilizes pre-trained stable diffusion models and their variants, with minor differences, such as the incorporation of Sdedit [5], being tricks to enhance generation quality. Thus, the manuscript fails to substantiate its claims and contributions convincingly.”**
> >
> > Neural network model structures, including image reconstruction strategies, are often inductive biased. On the one hand, researchers in the BCI community have been using spatial-temporal convolutional models to process EEG signals since EEGNet. On the other hand, the open source of StableDiffusion allows us to leverage models like CLIP that were trained with massive datasets as a teacher to guide the training of our brain models where we have a relative scarcity of data.
> >
> > Although it utilizes existing machine learning techniques , we demonstrate for the first time that EEG-based zero-shot visual decoding and reconstruction can be competitive with fMRI. Previous published work either focused on the EEG image decoding, or reconstructed images of known categories on small-scale and controversial datasets; or focused on the fMRI based image decoding and reconstruction, to achieve advanced performance on the fMRI datasets.
> >
> > Our work has introduced joint subject training for cross-subject evaluation, which is expected to solve the problem of decreased decoding performance due to subject differences when the amount of training data is sufficient. To the best of our knowledge, this is the first work to simultaneously achieve state-of-the-art performance on downstream zero-shot retrieval, classification, and reconstruction tasks on a dataset of the size of THINGS-EEG.
> >
> >
> > **Q3: “The processing of visual information in the brain involves multiple stages and typically takes around 100-150 milliseconds to reach higher visual areas where complex processing occurs. EEG signals at 50 milliseconds are likely still within the retina and optic nerve stages. Thus, the generation of images from 50ms EEG signals, as shown in Figure 7, contradicts established neuroscience principles. This suggests that the visual stimulus reconstruction framework heavily relies on the image generation model, which may have limited significance for the field of neuroscience.”**
> >
> > We have updated codes in the anonymous code link (https://anonymous.4open.science/r/Visual_Reconstruction-AC56). And we provided examples of reconstructing images with different random seeds for growing windows across time scales (README.md). From the two example stimulus images provided, the reconstructed images show uncertainty between 0 to 50 ms and 0 to 250 ms because the stimulus has just arrived at the primary visual cortex and it takes time for the brain to react and be captured, the reconstructed image during this period shows uncertainty (probably due to noise unrelated to the stimulus). Here, since the prior of natural images is added to the two-stage framework, high quality images can be reconstructed even from 0 to 50 ms and 0 to 250 ms. Then, with the accumulation of visual stimulus information, the semantics of the reconstructed images gradually become clear. After 500 ms, the images decoded by EEG tend to be stable, which means that there may be no new information added.
> >
> > Combined with Figure 7, the random seed image reconstruction example figure linked to the anonymous code we provided, and Figure 29 in Appendix H.1, we can see that these results just confirm the previous research and are consistent with the priors of neuroscience.

---

### Official Review · Reviewer_zVcP · 2024-07-12

**Soundness:** 3
**Presentation:** 3
**Contribution:** 3
**Rating:** 6
**Confidence:** 4

**Summary:**

The study proposes an end-to-end EEG\MEG-to-image reconstruction framework, consisting of a tailored brain encoder ATM to project neural signals into the shared subspace as the clip embedding, and a two-stage image generation block. The model achieves successful cross-subject EEG\MEG decoding and SOTA performance in classification, retrieval, and reconstruction.

**Strengths:**

1. The paper is well-organized and nicely written.
2. The experiments are comprehensive and convincing.

**Weaknesses:**

1. I think some parts of the model and implementation have not been clarified very well:
* I think it hasn’t been clarified in the main text whether the main results (starting from 3.2 to 3.5) show within-subject or cross-subject performance, unless I missed something. Adding some notices in figure\table captions or some summarizing sentences at the beginning of each section might be helpful.
* What model is inside the frozen “VAE image encoder” for low-level image generation in Figure 2? Also, what diffusion model is this study conditioning on? Is it built and retained from a pre-trained model (like stable diffusion), or was it trained from scratch by the authors?
* I think the authors didn’t introduce the “image2text” component (BLIP2), unless I missed it.
2. There are two recent EEG-to-image reconstruction works that this study has not discussed or compared. While it is understandable that the authors did not compare them, as they are still preprints and use different datasets, it might be beneficial to discuss them in the related work section given the limited literature in this field:
* Bai, Yunpeng, et al. "Dreamdiffusion: Generating high-quality images from brain eeg signals." arXiv preprint arXiv:2306.16934 (2023).
* Lan, Yu-Ting, et al. "Seeing through the brain: image reconstruction of visual perception from human brain signals." arXiv preprint arXiv:2308.02510 (2023).

For other minor comments please see Questions.

**Questions:**

1. For low-level metrics, in Mindeye, Ozcelik et al.’s, and other fMRI-to-image reconstruction works, they usually provide PixCorr values. I wonder why this metric wasn't included here.
2. For Figure 4, the plots are impressive, but it would be helpful to include numerical indications of the downstream task performances, such as 0.7 or 0.8. This would provide a clearer understanding of how well the model is performing.
3. Could the author elaborate more on the functioning of the shared token and subject tokens?

**Limitations:**

The authors have adequately discussed the limitations.

---

> ### Author Rebuttal · Authors · 2024-08-07
>
> Thank you for your careful review and comments. Below please find our point-to-point responses to your comments.
>
> **Q1. “I think it hasn’t been clarified in the main text whether....”**
>
> **Regarding 3.2 to 3.5:** We present the bar graphs of performance in the **Section 3.2 EEG Decoding Performance**, which are the absolute results of the within-subject. The detailed tables of retrieval performance are in **Appendix H.1**. In the **Section 3.3 Image Generation Performance**, we used the EEG data of subject 8 as a representative for image reconstruction and calculated mertics.
>
> There are statistical images reconstructed by a single subject in **Appendix G**. The experiments related to ‘Section 3.4 Temporal Analysis’ and ‘Section 3.5 Spatial Analysis’ were all conducted on the data of subject 8. We will follow your suggestion and add clear sentences in the titles of the figures and tables for explanation.
>
> **Q2. What model is inside the frozen “VAE image encoder” in Figure 2?**
>
> The details of this part are similar to [1]. We now make complementary explanations: The VAE model from the pre-trained StableDiffusion-v2.1-base is used for low-level image reconstruction, as shown in **Figure 2**. We use the latent obtained by VAE to align the latent of EEG and then reconstruct the low-level image from EEG. First, the preprosessed EEG passes through an MLP and upsampling CNN to obtain a 4x32x32 latent, and then passes through MSE loss, contrastive loss, reconstruction loss, and aligns the latent from the original image 3x256x256 after VAE encoding. We pass EEG latent through the VAE decoder to get a blurred 3x256x256 image, and then input the image into SDXL-turbo as low level guidance.
>
> **Q3. The authors didn’t introduce the “image2text” component (BLIP2).**
>
> Thank you for pointing out a typo, and we didn't make it clear here. The BLIP2 model [7] is indeed used in this work, but it is only used to extract text features from the original image for EEG classification. In the image reconstruction stage, to provide guidance from text features, we adopted the same approach as [5], using the GIT model [6] to obtain text descriptions from image embeddings in the reconstruction stage, which is then input into the SDXL-turbo decoder as a condition.
>
> **Q4. Why are these two papers not compared? [2][3]**
>
> In [2] and [3], both were conducted on a controversial dataset [4]. The experimental paradigm of this dataset was once accused of having a block design error, resulting in the contribution of decoding not entirely coming from the category itself. The experimental paradigm of the THINGS-EEG does not have this problem, because the training set of this dataset has 1654 categories and the test set has 200 categories, and the categories of its training set and test set are completely different.
>
> **Q5. Regarding the ‘PixCorr’ metric:**
>
> Due to the limited length of the table and the fact that the low-level metric already has SSIM, this is omitted in the manuscript. The performance on different benchmarks is summarized as follows:
> | Metric \ Method| Benchetrit Y et al. [9] (fMRI) | Fatma Ozcelik et al. [10] (fMRI) | Paolo Scotti et al. [1] (fMRI) | Benchetrit Y et al. [9] (MEG) | Ours (EEG) | Ours (MEG) |
> |-------------|-------------|-------------|-------------|-------------|-------------|-------------|
> | PixCorr   | 0.305 | 0.254 | 0.130 | 0.058 | **0.160** | **0.104** |
>
> **Q6. For Figure 4, the plots are impressive, but ....**
>
> Thank you for your suggestion, for **Figure 4**, our original intention was to make the radar chart as simple and intuitive as possible. Detailed performance tables for decoding and reconstruction are given in **Appendix H.1 Table 7** and in the rebuttal to the **reviewer (asin)** for more detailed results.
>
> **Q7. The functioning of the shared token and subject tokens.**
>
> Similar to [8], we leverage joint subject training to adapt to new subjects. Once a model is trained, it can be used for both reasoning about known subjects (subject-specific tokens) and reasoning about unknown subjects (shared tokens).
>
> **Reference**
>
> [1]Scotti P, Banerjee A, Goode J, et al. Reconstructing the mind's eye: fMRI-to-image with contrastive learning and diffusion priors[J]. Advances in Neural Information Processing Systems, 2024, 36.
>
> [2]Bai, Yunpeng, et al. "Dreamdiffusion: Generating high-quality images from brain eeg signals." arXiv preprint arXiv:2306.16934  (2023).
>
> [3]Lan, Yu-Ting, et al. "Seeing through the brain: image reconstruction of visual perception from human brain signals." arXiv preprint arXiv:2308.02510 (2023).
>
> [4]Spampinato, C.; Palazzo, S.; Kavasidis, I.; Giordano, D.; Souly, N.; and Shah, M. 2017. Deep learning human mind for automated visual classification. In Proceedings of the IEEE/CVF Conference on Computer Vision and Pattern Recognition, 6809–6817.
>
> [5]Ferrante M, Boccato T, Ozcelik F, et al. Multimodal decoding of human brain activity into images and text[C]//UniReps: the First Workshop on Unifying Representations in Neural Models. 2023.
>
> [6]Wang J, Yang Z, Hu X, et al. GIT: A Generative Image-to-text Transformer for Vision and Language[J]. Transactions on Machine Learning Research.
>
> [7]Li J, Li D, Savarese S, et al. Blip-2: Bootstrapping language-image pre-training with frozen image encoders and large language models[C]//International conference on machine learning. PMLR, 2023: 19730-19742.
>
> [8]Zhou Q, Du C, Wang S, et al. CLIP-MUSED: CLIP-Guided Multi-Subject Visual Neural Information Semantic Decoding[C]//The Twelfth International Conference on Learning Representations.
>
> [9]Benchetrit Y, Banville H, King J R. Brain decoding: toward real-time reconstruction of visual perception[C]//The Twelfth International Conference on Learning Representations.
>
> [10]Fatma Ozcelik and Rufin VanRullen. Natural scene reconstruction from fmri signals using generative latent 342 diffusion. Scientific Reports, 13(1):15666, 2023.

---

> > ### Comment · Reviewer_zVcP · 2024-08-11
> > **Thank you for the rebuttal**
> >
> > Thank you for the detailed explanations! I will maintain my current score.

---

> ### Author Response · Authors · 2024-08-12
> **Response to Reviewer zVcP**
>
> Thank you to the reviewer for the impressive questions and suggestions.
>
> We are very grateful to the reviewer for pointing out the technical details that were not clarified in our article. The content presented in this article is quite comprehensive and sufficient, but we sincerely hope the reviewer to pay further attention to our innovation and unique contributions:
>
> 1. Technically, we consider the positional relationship between channels and the modeling of the time dimension, so we introduce a channel-wise patch embedding and feedforward neural network in the time dimension (Figure 3). EEG data is different from fMRI, which requires sufficiently effective feature extraction and neuroscience-specific inductive paranoia to maximize the performance expectations of decoding - we have achieved state-of-the-art on different tasks (Figure 4). Finally, in the uploaded pdf file, we provide comprehensive evaluation results.
>
> 2. In terms of scientific insights, based on strong performance, we further provide analysis results of different dimensions such as time distribution (Figure 7), spatial distribution (Figure 8), representation distribution (Figure 11) and concept distribution (Figure 12), which strongly proves the effectiveness, scalability, interpretability and causality of our framework for EEG decoding and reconstruction.
>
> 3. In terms of potential impact, our work enhances the interpretability of the model on existing neural decoding tasks as much as possible, including decoding of cognitive concept foundations (Figure 12) and revealing neural mechanisms (Figure 8). Furthermore, we focus on the study of causal relationships. In order to understand the reasons for the high decoding accuracy, we conducted a number of spatial-temporal ablation experiments, compared and demonstrated in detail the corresponding viewpoints in EEG decoding and neuroscience. In addition, in order to prove the effectiveness of the results, future work will focus on the verification of real-world applications - we are collecting enough EEG data from different subjects for full parameter or efficient parameter fine tuning to further verify the effectiveness of our framework.
>
> 4. We have updated codes in the anonymous code link (https://anonymous.4open.science/r/Visual_Reconstruction-AC56), and all codes provided in the review area will be refined and be public after the anonymous phase. And we provided examples of reconstructing images with different random seeds for growing windows across time scales (README.md). From the two example stimulus images provided, the reconstructed images show uncertainty between 0 to 50 ms and 0 to 250 ms because the stimulus has just arrived at the primary visual cortex and it takes time for the brain to react and be captured, the reconstructed image during this period shows uncertainty (probably due to noise unrelated to the stimulus). Here, since the prior of natural images is added to the two-stage framework, high quality images can be reconstructed even from 0 to 50 ms and 0 to 250 ms. Then, with the accumulation of visual stimulus information, the semantics of the reconstructed images gradually become clear. After 500 ms, the images decoded by EEG tend to be stable, which means that there may be no new information added.

---

### Official Review · Reviewer_asin · 2024-07-14

**Soundness:** 3
**Presentation:** 3
**Contribution:** 3
**Rating:** 6
**Confidence:** 3

**Summary:**

This paper proposes a learning framework to decode images from EEG signals. It introduces a tailored brain encoder, the Adaptive Thinking Mapper, which projects neural signals to the clip embedding space. Subsequently, a two-stage image generation strategy is applied to produce images, progressing from blurry to high-quality reconstructions. This is an interesting and innovative work.

**Strengths:**

* This paper is easy to understand, and it is an interesting work exploring EEG to image decoding.

**Weaknesses:**

* Some experimental settings were not very clearly illustrated. For instance, in Section 3.1, it is mentioned that the experiments were trained on an EEG dataset and tested on an MEG dataset. How did you align the channel heterogeneity? Is it a zero-shot approach?

* Table 1 is a bit confusing to me. It would be helpful to clarify and explain the experimental settings of these additional datasets.

* Although EEG has a fast response, the design of the proposed framework does not seem to leverage motivations related to EEG data. It would be better to demonstrate the effectiveness of the proposed framework compared to existing fMRI-image decoding methodologies. For instance, you could replace the EEG encoder with an fMRI encoder and test whether the framework can outperform existing methodologies.

**Questions:**

* Is the spider plot in Figure 4 showing normalized performance? Please clarify.

* Please address my queries in the Weakness section:

**Limitations:**

I agree with the limitation regarding the cross-subject performance drop. Future work could consider incorporating existing cross-subject generalization efforts from the EEG domain to address this issue.

---

> ### Author Rebuttal · Authors · 2024-08-07
>
> Thank you for your time and thorough comments! Below please find our point-to-point responses to your comments.
>
> **Q1. How did you align the channel heterogeneity? Is it a zero-shot approach?**
>
> **Regarding Section 3.1:** ”To verify the versatility of ATM for embedding electrophysiological data, we tested it on MEG data modality using the THINGS-MEG dataset" :
>
> What we mean in the text is that in order to prove that our framework is universal for general embedding electrophysiological data, we also test the performance of our framework on the THINGS-MEG dataset. As can be seen in **Figure 4**, the MEG dataset and the EEG dataset are different evaluation systems because they are trained and evaluated separately. Therefore, the issue of channel heterogeneity is not involved here. Similar to the THINGS-EEG dataset, we also performed zero-shot testing on the THINGS-MEG dataset.
>
> **Q2. “It would be helpful to clarify and explain the experimental settings of these additional datasets. ”**
>
> **Regarding Table1:** In order to intuitively demonstrate the advantages of our framework, we compared the performance of our method with different methods on EEG, MEG and even fMRI dataset. Under the zero-shot setting, our framework makes the image reconstruction performance surpass the performance of [1] on MEG data, and is closer to the performance of current methods [1][2][3] on fMRI data, which highlights the superiority of our framework and makes image decoding and reconstruction more practical.
>
> **Q3. “It would be better to demonstrate the effectiveness of the proposed framework compared to existing fMRI-image decoding methodologies. ”**
>
> Please kindly see the relevant answer in **Global rebuttal: Q1 and Q2.** We believe that the differences in the characteristics of fMRI and EEG themselves are the premise that the existing fMRI decoding technology cannot be applied in actual brain-computer interface tasks. Therefore, it is necessary to design a framework that is suitable for EEG real-time decoding and online image reconstruction applications. For EEG data with low signal-to-noise ratio, non-stationarity, and possible time window offset, it is impossible for EEG to be as simple as fMRI encoder. It requires special design to meet the requirements of replacing manual features. In terms of the superiority of our framework, compared to fMRI-based decoding that can only be attempted on datasets, our proposed EEG encoder can directly process EEG signals and respond quickly to complete a series of decoding and image reconstruction tasks.
> | Dataset                                 |  PixCorr  |  SSIM   | AlexNet (2) | AlexNet (5) | Inception |   CLIP   |   SwAV   |
> | --------------------------------------- | :-------: | :-----: | :--------: | :--------: | :-------: | :------: | :------: |
> | NSD-fMRI (Benchetrit_2023)              |   0.305   |  0.366  |   0.962    |   0.977    |   0.910   |   0.917  |   0.410  |
> | THINGS-EEG (NICE, Song Y et al.)                       |   0.142   |  0.276  |   0.739    |   0.832    |   0.659   |   0.722  |   0.612  |
> | THINGS-EEG (EEGNetV4, Lawhern et al.)                   |   0.140   |  0.302  |   0.767    |   0.840    |   0.713   |   0.773  |   0.581  |
> | THINGS-MEG (Benchetrit_2023)            |   0.058   |  0.327  |   0.695    |   0.753    |   0.593   |   0.700  |   0.630  |
> | THINGS-MEG (averaged) (Benchetrit_2023) |   0.090   |  0.336  |   0.736    |   0.826    |   0.671   |   0.767  |   0.584  |
> | **THINGS-EEG (ATM-S, Ours)**                     |   **0.160**   |  **0.345**  |   **0.776**    |   **0.866**    |   **0.734**   |   **0.786**  |   **0.582**  |
> | **THINGS-MEG (ATM-S, Ours)**                      |   **0.104**   |  **0.340**  |   **0.613**   |   **0.672**    |   **0.619**   |   **0.603**  |   **0.651**  |
>
> In order to demonstrate the competitiveness of our framework, we used different neural encoder to test the reconstruction performance on the THINGS-EEG and THINGS-MEG dataset and the results are shown above.
>
> **Q4. Is the spider plot in Figure 4 showing normalized performance?**
>
> As shown in **Figure 4**, in each metric direction, we use the accuracy calculated by the best performing method as the benchmark and draw a normalized radar chart to more intuitively show the superiority of our method. Its absolute performance can be seen in **Table 7** in the **Appendix H.1**.
>
> **Reference**
>
> [1]Benchetrit Y, Banville H, King J R. Brain decoding: toward real-time reconstruction of visual perception[C]//The Twelfth International Conference on Learning Representations.
>
> [2]Fatma Ozcelik and Rufin VanRullen. Natural scene reconstruction from fmri signals using generative latent diffusion. Scientific Reports, 13(1):15666, 2023.
>
> [3]Paolo Scotti, Arpan Banerjee, Jessica Goode, et al. Reconstructing the mind’s eye: fmri-to-image with contrastive learning and diffusion priors. Advances in Neural Information Processing Systems, 36, 2024.
>
> [4]Takagi Y, Nishimoto S. High-resolution image reconstruction with latent diffusion models from human brain activity[C]//Proceedings of the IEEE/CVF Conference on Computer Vision and Pattern Recognition. 2023: 14453-14463.

---

### Author Rebuttal · Authors · 2024-08-07

Dear Area Chairs and Reviewers,

We express our profound gratitude for the comprehensive feedback and comments on our manuscript. This paper receives relatively serious differentiation of Weakly Accept, Weakly Accept, Reject, and Reject during the review period. We are excited about the consensus among the reviewers regarding the innovative (asin) and convincing (zVcP) impact of our work. However, we are saddened that two reviewers (4xL4 and gPm4) have some concerns about the novelty and ‘conflict ’ of our manuscript to neuroscience priors. So we hope to do our best to clarify what we did not explain clearly in the manuscript to allay the concerns of the reviewers (4xL4 and gPm4).
In the following , we provide a general response to several reviewers' questions and concerns.

**Q1. What is the motivation for this paper?**

With the increase of public data sets and the rapid development of generative models, visual decoding and reconstruction methods [1][2] based on fMRI datasets [3] have become very advanced, but few methods based on EEG datasets have been formally published in top conferences or journals [5][6]. fMRI has a natural advantage in spatial resolution, but for well-known reasons, fMRI-based work is almost impossible to apply in practice, and this have hindered the development of the brain-computer interface (BCI). This motivates us to propose a zero-shot visual decoding and reconstruction framework that can be proven effective on EEG datasets. Our manuscript provides more empirical guidance on practical BCI applications. We hope that the BCI and neuroscience communities pay more attention to the implementation of similar technologies and real data rather than overfitting only on existing fMRI datasets.

**Q2. What is the innovation of this paper? (How is it different from previous work?)**

**Our work is by no means a pile-up or improvement of existing work.** We propose a novel and feasible EEG-based zero-shot image decoding and reconstruction framework. Although it utilizes existing machine learning techniques , we demonstrate for the first time that EEG-based zero-shot visual decoding and reconstruction can be competitive with fMRI.
Previous published work either focused on the EEG image decoding [7], or reconstructed images of known categories on small-scale and controversial datasets [4]; or focused on the fMRI based image decoding and reconstruction, to achieve advanced performance on the fMRI datasets [3].
We believe that zero-shot visual reconstruction of EEG is essential, which is an important prerequisite for decoding imaginary images. The work in Song et al. [9] is that they only use Graph-attention or Self-attention modules, without any learnable token embedding strategies and Feed Forward Layer, and even position encoding schemes. In addition, our method provides a plug-and-play spatial-temporal convolution module, and we prove that even if EEGNetV4 is used as a convolution module, the performance is still robust. Our reconstruction scheme is also different from Bencherit Y el al. [8], which only used the method in MindEye [1] in their paper for MEG data, and the performance of our two-stage image reconstruction framework is far behind.

**Q3. The conclusions of this article seem to violate neuroscientific priors?**

The results of this paper are exactly mutually confirmed to the conclusions of previous papers and are consistent with neuroscience priors [7][8]. According to our results in **Appendix H.1 Accuracy for time windows**, **Figure 29** shows that for all embeddings, a clear peak can be observed for windows ending around 200-250 ms after image onset. Comparing **Figure 28** and **Figure 29**, we can see that, unlike [8], our time window results on THINGS-EEG dataset do not have a second peak, which may be mainly affected by the experimental paradigm. We can see that in the first 50ms-200ms, the image reconstructed at 50 ms is completely messy and has no semantics; the semantics of the reconstructed image after 250 ms is basically correct and gradually stabilizes, and after 500ms, due to the lack of additional visual response, the content of the image reconstruction is more stable. Similar results are also shown in **Figure 4 A of [8]**: their method can also reconstruct high-quality images at 100ms. **This just shows that our reconstruction results are in line with the neuroscience prior.**
However, this does not mean that the EEG data after the absence of visual response (200ms) loses its contribution to decoding, because the processing of high-level visual features (corresponding to the visual features of CLIP) may be involved over time.


**Reference**

[1]Scotti P, Banerjee A, Goode J, et al. Reconstructing the mind's eye: fMRI-to-image with contrastive learning and diffusion priors[J]. Advances in Neural Information Processing Systems, 2024, 36.

[2]Xia W, de Charette R, Öztireli C, et al. UMBRAE: Unified Multimodal Decoding of Brain Signals[J]. arXiv preprint arXiv:2404.07202, 2024.

[3]Allen, E. J. et al. A massive 7t fmri dataset to bridge cognitive neuroscience and artificial intelligence. Nat. Neurosci. 25, 116–126 (2022).

[4]Kavasidis I, Palazzo S, Spampinato C, et al. Brain2image: Converting brain signals into images[C]//Proceedings of the 25th ACM international conference on Multimedia. 2017: 1809-1817.

[5]Bai, Yunpeng, et al. "Dreamdiffusion: Generating high-quality images from brain eeg signals." arXiv preprint arXiv:2306.16934 (2023).

[6]Lan, Yu-Ting, et al. "Seeing through the brain: image reconstruction of visual perception from human brain signals." arXiv preprint arXiv:2308.02510 (2023).

[7]Song Y, Liu B, Li X, et al. Decoding Natural Images from EEG for Object Recognition[C]//The Twelfth International Conference on Learning Representations.

[8]Benchetrit Y, Banville H, King J R. Brain decoding: toward real-time reconstruction of visual perception[C]//The Twelfth International Conference on Learning Representations.

---

### Author Response · Authors · 2024-08-12
**Sincere Request for Feedback on Response Clarifications and Explanations**

Dear Chairs and Reviewers,

Thanks for the insightful questions and suggestions to our manuscript.

Could you please provide feedback on our responses to your questions? We are sincerely eager to know if our answers have addressed your concerns. Additionally, are there any further comments or insights you might have regarding our work?

Your input is crucial for us to enhance the quality of our submission, and we greatly appreciate your guidance and time.

Best wishes,

Paper Authors

---

### Comment · Area_Chair_YBnx · 2024-08-14
**please respond to the extensive rebuttal**

Reviewer 4xL4 please respond to the extensive rebuttal provided by the authors.

---

### Public Comment · ~Yun_Wang9 · 2024-12-26
**Overlapped time window between trials?**

The EEG dataset presented stimuli for 100ms with a SOA of 200ms. Why do the trials use the time window from 0-1000ms? Will this cause overlapped signals between trials?

---

### Decision · Program_Chairs · 2024-09-25

**Decision:**

Accept (poster)

**Comment:**

This paper demonstrates visual decoding and reconstruction of viewed
images from recorded EEG using the THINGS EEG(2) dataset.  They
demonstrate their algorithm on image retrieval, category
classification, and image reconstruction tasks.  Results are
impressive and exciting.  The results (timing, most relevant
electrodes) are consistent with the Neuroscience literature, but the
ability to do this with EEG is surprising.

The biggest issue with this paper, as raised by one reviewer is
measuring performance as maximal on test data.  While the author responded
that it should not matter as all methods are tested similarly, this is
not true as methods with higher variance are more benefitted by this
approach.  It is well known that parameters must be fit on data
separate from the test data.  If this paper is accepted, this will
have to be changed in the camera ready.	 From the plots, it appears that
this will only change the numbers slightly, and	so I recommend acceptance.

Minor:
It appears there might be a typo in the	supplemental text.  On line
432, it is mentioned that "we averaged across the 4 EEG trials from
the same image in the test set" but the test set has 80 repeats and
later on line 513, they say they used an "averaging technique on 80 repeated instances
within the test set".

Finally, there is contemporaneous work for reconstruction from EEG
here https://arxiv.org/abs/2404.01250